# UnHype: CLIP-Guided Hypernetworks for Dynamic LoRA Unlearning

**Piotr Wójcik** [* 1] **Maksym Petrenko** [* 2] **Wojciech Gromski** [* 3 4] **Przemysław Spurek** [2 4] **Maciej Zieba** [3 5]

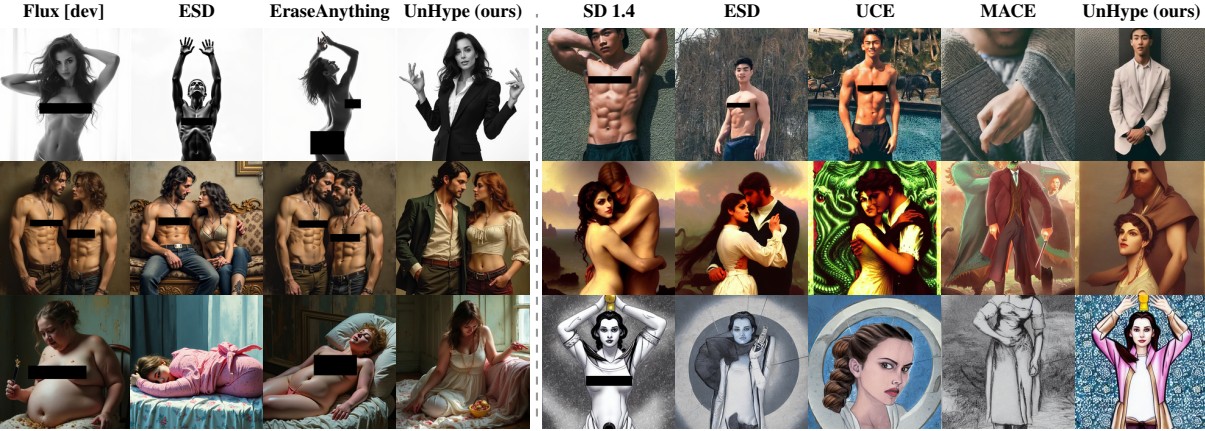

*Figure 1. Left*: Comparative evaluation of explicit content erasure on the Flux architecture. We display the output of the original model alongside results from existing baseline methods and **UnHype**. *Right*: A parallel comparison conducted on Stable Diffusion, contrasting the original model's generation against competing approaches and our proposed framework.

## Abstract

Recent advances in large-scale diffusion models have intensified concerns about their potential misuse, particularly in generating realistic yet harmful or socially disruptive content. This challenge has spurred growing interest in effective machine unlearning, the process of selectively removing specific knowledge or concepts from a model without compromising its overall generative capabilities. Among various approaches, Low-Rank Adaptation (LoRA) has emerged as an effective and efficient method for fine-tuning models toward targeted unlearning. However, LoRA-based methods often exhibit limited adaptability to concept semantics and struggle to balance removing closely related concepts with maintaining generalization across broader meanings. Moreover, these methods face scalability challenges

when multiple concepts must be erased simultaneously. To address these limitations, we introduce UnHype, a framework that incorporates hypernetworks into single- and multi-concept LoRA training. The proposed architecture can be directly plugged into Stable Diffusion as well as modern flow-based text-to-image models, where it demonstrates stable training behavior and effective concept control. During inference, the hypernetwork dynamically generates adaptive LoRA weights based on the CLIP embedding, enabling more context-aware, scalable unlearning. We evaluate UnHype across several challenging tasks, including object erasure, celebrity erasure, and explicit content removal, demonstrating its effectiveness and versatility. See the code on GitHub.

## 1. Introduction

Machine unlearning has become an increasingly important task in modern machine learning, particularly as large-scale generative models continue to evolve and permeate real-world applications. The ability to selectively remove knowledge or concepts from a trained model is crucial for addressing ethical, legal, and safety concerns, such as enforcing data privacy regulations and preventing the generation of harmful or malicious content. Effective unlearning aims

---

[*]Equal contribution [1]CMMC Center for Molecular Medicine Cologne, University of Cologne [2]Faculty of Mathematics and Computer Science, Jagiellonian University, Krakow, Poland [3]Wrocław University of Science and Technology [4]IDEAS Research Institute [5]Tooploox. Correspondence to: Piotr Wójcik <piotr.m.wojcik@gmail.com>.

*Proceedings of the 43rd International Conference on Machine Learning*, Seoul, South Korea. PMLR 306, 2026. Copyright 2026 by the author(s).

to erase specific information while preserving the model's overall performance and generalization capabilities, making it a key challenge in the era of powerful diffusion models.

Recent progress in text-to-image diffusion models has demonstrated remarkable capabilities in generating highly realistic and semantically coherent images. However, these advancements have also heightened concerns regarding misuse, including the creation of explicit, biased, or socially disruptive imagery. Consequently, numerous efforts have been made to develop mechanisms that can selectively unlearn undesirable content. Among these, Low-Rank Adaptation (LoRA) (Hu et al., 2022) has emerged as a promising and efficient approach for parameter-efficient fine-tuning. By introducing low-rank matrices into the attention and feed-forward layers of pre-trained diffusion models, LoRA enables targeted adaptation without full retraining, making it well-suited for concept-level unlearning. Several recent studies (Lu et al., 2024; Polowczyk et al., 2025) have leveraged LoRA to suppress specific concepts or objects, achieving notable success in reducing unwanted generations while maintaining visual fidelity.

Despite their effectiveness, existing LoRA-based unlearning methods exhibit several key limitations. First, they apply a global, static weight modification — once merged, the adapter affects every forward pass regardless of the input prompt, which can lead to overly broad forgetting that degrades semantically adjacent concepts. Second, this rigid structure limits flexibility with context-dependent or compositional prompts. Third, while the per-concept training cost of a single LoRA is modest, scalability becomes a practical bottleneck when many concepts must be erased simultaneously, as each requires a separate run with its own hyperparameter tuning and checkpointing.

To overcome these challenges, we propose UnHype, a novel unlearning framework that integrates hypernetworks with LoRA for both single- and multi-concept forgetting. In our approach, the hypernetwork dynamically generates LoRA weights conditioned on the CLIP embedding of the input concept. This design allows the model to produce adaptive, context-aware LoRA updates that generalize across semantically related concepts while maintaining precise control over what is removed. During inference, UnHype efficiently adjusts its unlearning behavior without retraining or manual intervention, enabling scalable and flexible concept erasure. Experimental results across multiple benchmarks, including object removal, celebrity unlearning, and explicit content suppression (Figure 1), demonstrate that UnHype achieves superior balance between effective forgetting and preservation of generative quality, establishing a new direction for adaptive machine unlearning in diffusion models.

The contributions of this work are summarized as follows:

- A novel, context-aware machine unlearning framework that combines hypernetworks with Low-Rank Adaptation (LoRA) for use in both Stable Diffusion and flow-based models like Flux. The system dynamically generates adaptive weights conditioned on CLIP embeddings to enable semantically guided forgetting without further retraining.

- Unlike previous methods that require separate LoRA fine-tuning for every target, this design supports simultaneous unlearning across multiple concepts within a single training run, significantly lowering computational costs in the multi-concept regime.

- Extensive experiments in object, celebrity, and explicit content erasure show that UnHype offers superior trade-offs between effective forgetting, visual fidelity, and generalization compared to existing baselines.

## 2. Related Work

The concept of machine unlearning was formally introduced by (Kurmanji et al., 2023) in the context of data deletion and privacy, emphasizing the challenge of removing the influence of particular training examples from a learned model. The straightforward solution, which consists of refining the training dataset and retraining the model, is computationally demanding and often impractical for large-scale systems (Carlini et al., 2022; O'Connor, 2022). Alternative strategies such as post-generation filtering or inference-time control have also been explored, but these methods are typically unreliable, as users can often bypass such restrictions (Rando et al., 2022; Schramowski et al., 2023).

In the domain of diffusion models, recent research has focused on designing more efficient unlearning mechanisms. Several studies employ fine-tuning procedures to suppress specific concepts or visual content. For instance, EDiff (Wu et al., 2024) introduces a bi-level optimization framework, while ESD (Gandikota et al., 2023) leverages a modified classifier-free guidance approach with negative prompts. FMN (Zhang et al., 2024a) proposes a re-steering loss applied to selected attention layers, thereby reducing activations related to unwanted content. Other works, including SalUn (Fan et al., 2023) and SHS (Wu & Harandi, 2024), adapt model parameters through saliency or sensitivity analysis to identify and adjust weights responsible for undesired concepts. SEMU (Sendera et al., 2025) employs Singular Value Decomposition (SVD) to construct a low-dimensional subspace that enables selective forgetting. Similarly, SA (Heng & Soh, 2023) and CA (Kumari et al., 2023) replace the distribution of unwanted features with surrogate or anchor representations, while SPM (Lyu et al., 2024) integrates lightweight structural adapters throughout the network to block the propagation of forbidden concepts. More

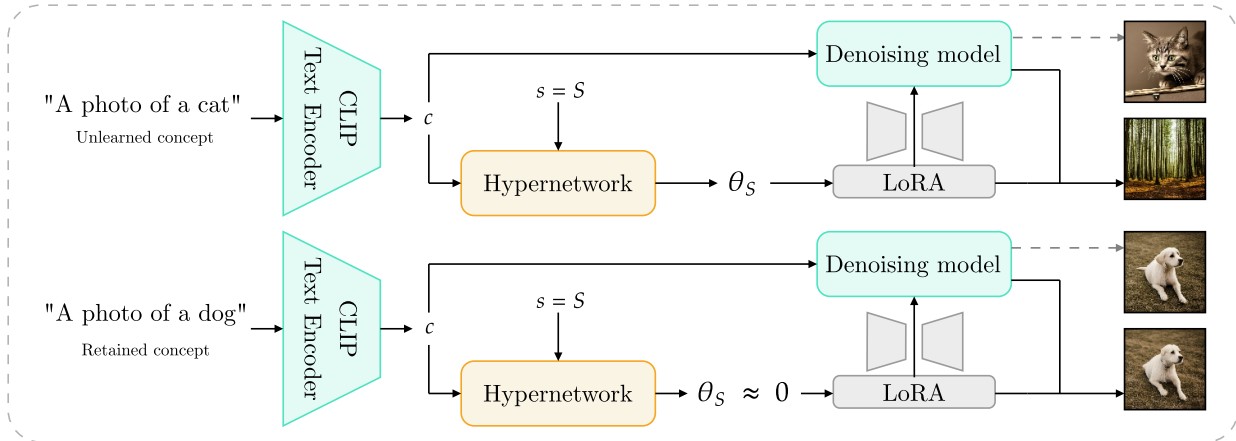

*Figure 2.* **Overview of the inference in UnHype.** The top part shows how the model handles an unlearned concept ("a photo of a cat"). The text embedding $c$ is fed into a hypernetwork that generates concept-specific LoRA parameters $\theta_S$. These parameters modify the denoising model to suppress the forbidden concept, producing an alternative image (a forest) instead. The bottom part shows a retained concept ("a photo of a dog"). In this case, the hypernetwork generates LoRA parameters close to zero ($\theta_S \approx 0$), which have a negligible effect on the denoising model, allowing it to generate the dog image as usual.

interpretable approaches, such as SAeUron (Cywiński & Deja, 2025), use sparse autoencoders to localize and remove concept-specific activations, achieving effective forgetting with minimal impact on generation quality and strong robustness against adversarial prompts.

A growing body of work explores *parameter-efficient* unlearning strategies. Low-Rank Adaptation (LoRA) (Hu et al., 2022), initially proposed for adding new concepts to text-to-image diffusion models, has been repurposed for selective forgetting. MACE (Lu et al., 2024) exemplifies this trend by combining two LoRA-based components: one dedicated to removing residual associations and another that directly erases the target concept. This method employs segmentation maps from Grounded-SAM (Liu et al., 2024) to spatially localize and suppress attention activations, although it depends on external segmentation tools and specialized adapter configurations.

Building upon this foundation, UnGuide (Polowczyk et al., 2025) extends LoRA-based unlearning by integrating dynamic inference-time guidance. The method learns LoRA modules that encode unwanted concepts and applies them selectively during the denoising process to control when and how suppression occurs. This hybrid design improves the balance between erasure precision and the preservation of general generation quality.

Overall, prior work indicates that modular, lightweight interventions, particularly LoRA and related adapter-based techniques, are effective for concept-level unlearning in diffusion models. However, most approaches either rely on external localization tools or face trade-offs between for-

getting accuracy and fidelity. These limitations motivate further exploration of adaptive, hypernetwork-driven LoRA mechanisms for efficient, interpretable, and controllable machine unlearning.

## 3. Background

**Unlearning** is the deliberate process of removing or suppressing specific knowledge, concepts, or associations from a trained machine learning model. Within the scope of generative models, unlearning seeks to disable the model's capacity to recognize, reproduce, or generate a particular concept, feature, or category (for instance, a specific individual, object class, or artistic style), while preserving its overall functionality and performance on other tasks. This process is often driven by ethical, legal, or privacy requirements, such as eliminating unauthorized content, addressing societal biases, or fulfilling data deletion requests. Unlike conventional retraining or fine-tuning, which augment a model with new information, unlearning selectively erases targeted knowledge to minimize its influence, striving to maintain the integrity of unrelated representations and the model's general abilities.

**Foundation Diffusion Models** such as Stable Diffusion (Rombach et al., 2022) and FLUX are conditional Latent Diffusion Models (LDMs) comprising three main components: a text encoder $\mathcal{T}$, a denoising model $\mathcal{U}_\theta$ parameterized by $\theta$, and a pretrained variational autoencoder (VAE) (Kingma & Welling, 2013) with encoder $\mathcal{E}$ and decoder $\mathcal{D}$. To achieve computational efficiency, LDMs operate in a compressed latent space instead of the high-dimensional

pixel space.

The diffusion model in the latent space operates by modeling a Markov chain of successive noise addition and denoising steps. These models involve a forward process, where noise is gradually added to the encoded image, and a reverse process, where the model learns to remove noise, generating a latent representation of high-quality samples from a noise vector.

The forward process is defined as a series of Gaussian noise steps applied to the encoded image $z_0 = \mathcal{E}(x_0)$, transforming it into increasingly noisy versions $z_t$ as $t$ progresses from 0 to $T$. This process can be described as:

$$q(z_t|z_{t-1}) = \mathcal{N}(z_t; \sqrt{1 - \beta_t}z_{t-1}, \beta_t\mathbf{I}), \qquad (1)$$

where $\{\beta_t \in (0,1)\}_{t=1}^T$ is a noise schedule parameter that controls the level of noise added at step $t$.

The reverse process, modeled by the diffusion model, attempts to reconstruct $z_0$ from a noisy $z_T$ by progressively denoising it. The goal of training is to learn a model $p_\theta(z_{t-1}|z_t)$ that can reverse the noise process. In practice, the model is often parameterized as $\epsilon_\theta(z_t, t)$ and trained to predict the real noise applied to $z_0$, following equation (1). Once the final clean latent vector $z_0$ is obtained, the VAE's decoder, $\mathcal{D}$, transforms it into the final pixel-space image $x_0$. The conditioning factor $c$ is represented by the encoded text prompt and injected into $\epsilon_\theta(z_t, t, c)$ to guide the diffusion process to be consistent with the prompt.

Classifier-Free Guidance (CFG) (Ho & Salimans, 2022) is a technique employed in diffusion models to enhance control over the generative process. It has shown significant effectiveness in boosting the quality of generated outputs across tasks such as image and text generation. For a noisy sample $z_t$, this guidance is implemented by interpolating between these conditional and unconditional predictions as follows:

$$\epsilon_{\theta*}^{\text{CFG}}(z_t, t, c) = (1+w)\epsilon_{\theta*}(z_t, t, c) - w \cdot \epsilon_{\theta*}(z_t, t, c_0), \ (2)$$

where $\epsilon_\theta(z_t, t, c_0)$ represents the model's prediction of the noise for $x_t$ in the unconditional case (by providing an empty prompt $c_0$), while $\epsilon_\theta(z_t, t, c)$ denotes the noise prediction when conditioned on $c$. The parameter $w$ serves as the guidance scale, adjusting the extent to which the conditional information $c$ influences the generated output.

**Low-Rank Adaptation (LoRA)** (Hu et al., 2022) is an efficient parameter-tuning technique that adapts large, pre-trained models without modifying all of their original weights. Instead of fine-tuning the entire set of parameters, LoRA injects smaller, trainable low-rank matrices into the model's layers. The original model weights ($W$) are kept frozen, and the training process only learns the parameters of these small, rank-constrained modifications ($\Delta W$).

This approach drastically reduces both the computational cost and the memory required for training. This efficiency is achieved by representing the weight update ($\Delta W$) as the product of two low-rank matrices

$$W' = W + \alpha \cdot \Delta W = W + \alpha \cdot BA, \qquad (3)$$

where $A \in \mathbb{R}^{d \times r}$ and $B \in \mathbb{R}^{r \times k}$, with $d$ and $k$ being the dimensions of the original weight matrix $W$ and $r \ll \min(d,k)$. $\alpha$ is a scaling factor that controls the magnitude of the change. This method allows for efficient adaptation while retaining the model's original expressive capacity.

While Low-Rank Adaptation (LoRA) was initially developed for adding new concepts to Text-to-Image (T2I) models, recent work, such as MACE (Lu et al., 2024), has demonstrated its effectiveness for the opposite task: unlearning or removing specific target information.

**Hypernetworks** (Ha et al., 2016) are neural models that generate weights for another target network with the objective of solving a specific task. This approach results in a reduction of the number of trainable parameters compared to traditional methodologies that integrate supplementary information into the target model via a single embedding. A notable reduction in the size of the target model is achievable because it does not share global weights. Instead, these weights are provided by the hypernetwork. Most recent models utilize hypernetworks (Zieba et al., 2024; Ruiz et al., 2024) to predict LoRA parameters for image-to-image generation tasks.

## 4. Our Approach

In this section, we introduce UnHype, a novel framework for amortized machine unlearning in diffusion models. Our approach can be easily adapted to various types of conditional LDMs to which LoRA fine-tuning can be applied. An overview of our approach is illustrated in Figure 2. The conditioning prompt is processed by the CLIP Text Encoder to obtain its embedded representation, denoted as $c$. This representation is then injected into the denoising model in the conditional LDM.

Additionally, $c$ is used as input to the hypernetwork $H_\phi$, which predicts the LoRA parameters. The model is trained in such a way that if the input $c$ corresponds to a forbidden concept, the hypernetwork predicts LoRA parameters that suppress the generation of images containing that concept, effectively unlearning it and producing an alternative image instead. For other unrelated concepts, the hypernetwork predicts LoRA parameters close to zero, indicating that they have a negligible effect on the generation process, allowing the model to behave similarly to how it did before unlearning.

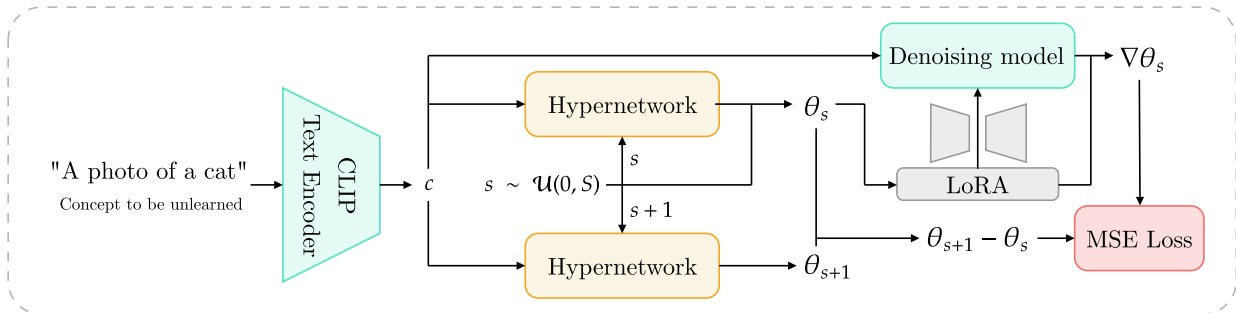

*Figure 3.* **Overview of the removal loss in UnHype.** The hypernetwork is queried at two consecutive steps, $s$ and $s+1$, to predict LoRA weights $\theta_s$ and $\theta_{s+1}$. The difference between these weights, $\theta_{s+1} - \theta_s$, forms the predicted step. Simultaneously, the target step of the task loss, $\Delta\theta_{\text{task}}$, is computed according to Equation (5). The removal loss is the MSE Loss between the predicted step and the target step, forcing the hypernetwork's trajectory to match the gradient field of the unlearning task.

While the per-step cost of LoRA fine-tuning is individually modest, per-concept approaches become a practical bottleneck in the multi-concept regime: erasing 100 celebrities requires 100 independent training runs with separate hyperparameter tuning and checkpointing (Ruiz et al., 2023; Gal et al., 2022). For single- or few-concept erasure, the training cost of UnHype is comparable to standard LoRA fine-tuning; the efficiency advantage manifests specifically through amortization over many concepts. We address this by reframing unlearning from a static fine-tuning task into a dynamic, amortized generation process. Instead of training a separate LoRA adapter for each concept, we generate it on-the-fly using a single, unified model. In our framework, the hypernetwork $H_\phi$ produces the LoRA weights for the diffusion model, a strategy that has been shown to be highly parameter-efficient for multi-task adaptation (Mahabadi et al., 2021). By conditioning on text embeddings, the hypernetwork generalizes beyond the observed training data, capturing synonyms and semantically related concepts that were not explicitly encountered during training.

### 4.1. UnHype: Unlearning with a Hypernet Field

One of the central challenges in our framework is designing and training a hypernetwork that predicts LoRA parameters in the desired manner. A naive training strategy introduces a fundamental obstacle: to learn a mapping of the form $H_\phi(c) \to \theta_c$ (where $\theta_c$ denotes the LoRA weights corresponding to the concept $c$), one would require a dataset of paired (concept, target-LoRA) examples. Constructing such a dataset would itself necessitate fine-tuning and storing a separate LoRA module for every concept of interest, precisely the static, per-concept bottleneck our method seeks to eliminate.

To circumvent this limitation, we draw inspiration from Hypernet Fields (Hedlin et al., 2025). Rather than learning

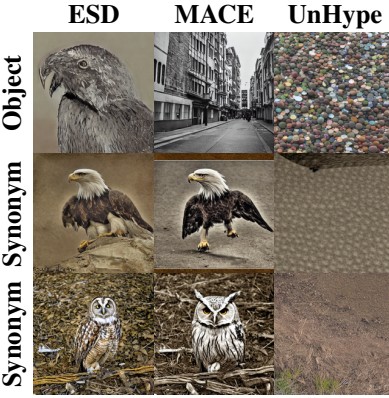

*Figure 4.* Qualitative comparison showing object erasure results on Stable Diffusion, where the concept *bird* is mapped to a neutral concept.

a direct mapping from a conditioning signal $c$ to a fixed parameter vector $\theta_S$, Hypernet Fields proposes modeling the entire optimization trajectory that leads to $\theta_S$, where $S$ denotes the last step of the unlearning trajectory. This is accomplished by augmenting the hypernetwork input with a continuous timestep variable $s$. The hypernetwork, therefore, outputs $\theta_s$ according to the mapping $(c, s) \to \theta_s$, where $\theta_s$ represents the parameter state at optimization step $s$. As (Hedlin et al., 2025) demonstrated, this formulation is crucial because it allows the hypernetwork to be trained by supervising its local gradients ($\nabla_s H_\phi$) with the gradients of a task loss ($\nabla_\theta \mathcal{L}$), completely eliminating the need for pre-computed final weights $\theta_S$.

As a consequence, we utilize a hypernetwork $H_\phi$ implemented as a Multi-Layer Perceptron (MLP) that dynamically generates the full set of LoRA weights $\theta_s$ for all target modules. In Stable Diffusion, we apply LoRA to the cross-attention mechanisms responsible for conditioning image generation on text prompts. For Flux, we modify

the corresponding value projection and output transformation components. This generation is conditioned on two inputs, $\theta_s = H_\phi(c, s)$, where $c$ is the 768-dimensional CLIP text embedding (Radford et al., 2021) of a concept, and $s \in [0, S]$ is the step of unlearning. This use of a hypernetwork to model a continuous, semantically-driven process is philosophically similar to work on modeling 3D shapes (Sitzmann et al., 2020) or image recontextualization (Zieba et al., 2024), and is a direct application of the gradient-matching principle from (Hedlin et al., 2025).

### 4.2. Training objective

To train $H_\phi$ to have this behavior, we must train it to follow the optimization path of an unlearning task *without* pre-computing that path. We formulate a training objective $\mathcal{L}_{final}$ as a weighted sum of two components: a "removal" loss that performs the unlearning and a "retention" loss that prevents catastrophic forgetting:

$$\mathcal{L}_{final} = \lambda_{remove} \cdot \mathcal{L}_{remove} + \lambda_{retain} \cdot \mathcal{L}_{retain}$$

**Unlearning Task Loss ($\mathcal{L}_{task}$)** First, we define the "task" we want the model to learn. We adapt the task loss from UnGuide, which recasts unlearning as a guided regression problem. The goal is to produce LoRA weights $\theta_s$ that, when added to the base model $\theta^*$, steer the denoising model prediction $\epsilon_{\theta_s+\theta^*}$ away from a "forget" concept $c$ and towards a "mapping" concept $c_m$ (e.g., "a cat" $\rightarrow$ "a forest"). The "steered" target prediction $\epsilon_{target}$ is defined as a linear combination of the *base* model's predictions:

$$\epsilon_{target} = \epsilon_{\theta^*}(z_t, t, c_m) - \gamma(\epsilon_{\theta^*}(z_t, t, c) - \epsilon_{\theta^*}(z_t, t, c_m)),$$

where $\gamma$ controls the degree to which the model is repelled from $c$ in favor of $c_m$.

The task loss $\mathcal{L}_{task}$ is then the Mean Squared Error between our *adapted* model's prediction and this target:

$$\mathcal{L}_{task} = \mathbb{E}_{z_t,t,c}[||\epsilon_{\theta_s+\theta^*}(z_t, t, c) - \epsilon_{target}||_2^2]. \quad (4)$$

**Removal Loss ($\mathcal{L}_{remove}$)** This loss implements the Hypernet Field gradient-matching principle. At each training step, we sample a "forget" concept $c$ and a random unlearning step $s \sim \mathcal{U}(0, S)$. We then enforce that the hypernetwork's numerical gradient, or "predicted step," $\Delta\theta_{pred}$, matches the analytical gradient of the task loss, or "target step," $\Delta\theta_{task}$. The overview of this principle is shown in Figure 3.

The **target step** is a standard SGD update for our task:

$$\Delta\theta_{task} = -\eta\nabla_{\theta_s}\mathcal{L}_{task} \quad (5)$$

This is the gradient of our unlearning task loss (Eq. 4) with respect to the *current* LoRA weights $\theta_s = H_\phi(c, s)$, scaled

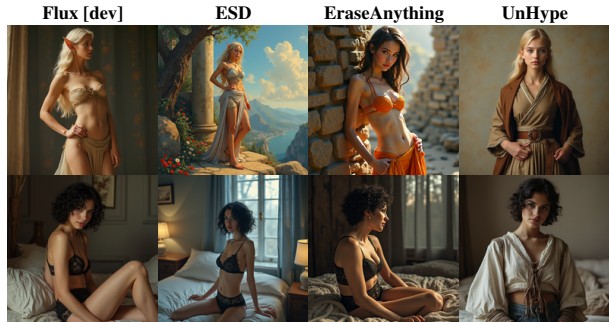

*Figure 5.* Qualitative comparison showing nudity erasure results on Flux. Prompts sampled from the I2P dataset.

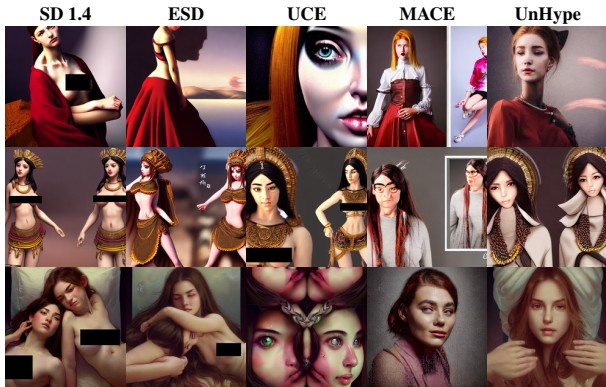

*Figure 6.* Qualitative comparison showing nudity erasure results on Stable Diffusion. Prompts sampled from the I2P dataset.

by a simulated learning rate $\eta$. The **predicted step** is the hypernetwork's own trajectory gradient:

$$\Delta\theta_{pred} = H_\phi(c, s + 1) - H_\phi(c, s)$$

The removal loss is the MSE between these two vectors, forcing the hypernetwork's trajectory to align with the gradient field of the unlearning task:

$$\mathcal{L}_{remove} = ||\Delta\theta_{pred} - \Delta\theta_{task}||_2^2$$
$$= ||(H_\phi(c, s + 1) - H_\phi(c, s)) + \eta\nabla_{\theta_s}\mathcal{L}_{task}||_2^2$$

**Retention Loss ($\mathcal{L}_{retain}$)** This loss enforces the "semantic switch" behavior and prevents catastrophic forgetting. When $H_\phi$ is conditioned on a "retain" concept $c_{retain}$, it should produce null weights for all steps $s$. We implement this by penalizing any deviation from the initial, zero-weight state $\theta_0 = H_\phi(c_{retain}, 0)$:

$$\mathcal{L}_{retain} = \mathbb{E}_{c_{retain},s}[||H_\phi(c_{retain}, s) - H_\phi(c_{retain}, 0)||_2^2]$$

This simple loss effectively trains the hypernetwork to output $\theta_s \approx 0$ when given a non-target concept, preserving the model's general capabilities.

In essence, UnHype successfully transforms the static, per-concept fine-tuning problem of unlearning into a dynamic,

| Method | Airplane Erased | | | | Ship Erased | | | | Bird Erased | | | | Average | | | |
|---|---|---|---|---|---|---|---|---|---|---|---|---|---|---|---|---|
| | $\text{Acc}_e \downarrow$ | $\text{Acc}_s \uparrow$ | $\text{Acc}_g \downarrow$ | $\text{H}_o \uparrow$ | $\text{Acc}_e \downarrow$ | $\text{Acc}_s \uparrow$ | $\text{Acc}_g \downarrow$ | $\text{H}_o \uparrow$ | $\text{Acc}_e \downarrow$ | $\text{Acc}_s \uparrow$ | $\text{Acc}_g \downarrow$ | $\text{H}_o \uparrow$ | $\text{Acc}_e \downarrow$ | $\text{Acc}_s \uparrow$ | $\text{Acc}_g \downarrow$ | $\text{H}_o \uparrow$ |
| FMN | 96.76 | 98.32 | 94.15 | 6.13 | 97.97 | 98.21 | 96.75 | 3.70 | 99.46 | 98.13 | 96.75 | 1.38 | 98.06 | 98.22 | 95.88 | 3.74 |
| AC | 96.24 | 98.55 | 93.35 | 6.11 | 98.18 | 98.50 | 77.47 | 4.97 | 99.55 | 98.53 | 94.57 | 1.24 | 97.99 | 98.53 | 88.46 | 4.11 |
| UCE | 40.32 | 98.79 | 49.83 | 64.09 | 6.13 | 98.41 | 21.44 | 89.44 | 10.71 | 98.35 | 15.97 | 90.18 | 19.05 | 98.52 | 29.08 | 81.24 |
| SLD-M | 91.37 | 98.86 | 89.26 | 13.69 | 89.24 | 98.56 | 41.02 | 24.99 | 80.72 | 98.39 | 85.00 | 23.31 | 87.11 | 98.60 | 71.76 | 20.66 |
| ESD-x | 33.11 | 97.15 | 32.28 | 74.98 | 33.35 | 97.93 | 34.78 | 73.99 | 18.57 | 97.24 | 40.55 | 76.17 | 28.34 | 97.44 | 35.87 | 75.05 |
| ESD-u | 7.38 | 85.48 | 5.92 | 90.57 | 18.38 | 94.32 | 15.93 | 86.33 | 13.17 | 86.17 | 20.65 | 83.98 | 12.98 | 88.66 | 14.17 | 86.96 |
| MACE | 9.06 | 95.39 | 10.03 | 92.03 | 8.49 | 97.35 | 10.53 | 92.61 | 9.88 | 97.45 | 15.48 | 90.39 | 9.14 | 96.73 | 12.01 | 91.68 |
| **UnHype** | 6.07 | 98.71 | 8.59 | **94.59** | 5.30 | 91.30 | 2.47 | **94.44** | 8.46 | 98.59 | 11.94 | **92.52** | 6.61 | 96.20 | 7.67 | **93.85** |
| SD v1.4 | 96.06 | 98.92 | 95.08 | - | 98.64 | 98.63 | 64.16 | - | 99.72 | 98.51 | 95.45 | - | 98.14 | 98.69 | 84.90 | - |

*Table 1.* **Evaluation of erasing specific classes.** Primary metrics: $\text{Acc}_e$, $\text{Acc}_s$, $\text{Acc}_g$; composite metric $\text{H}_o$. The final column represents the arithmetic mean of the metrics for these three classes. All values are percentages.

amortized generation task. By combining the gradient-matching principle of Hypernet Fields (Hedlin et al., 2025) with the targeted task loss from UnGuide, we train a single hypernetwork $H_\phi$. This network functions as a generative model for "unlearning adapters," capable of producing specialized LoRA weights for any arbitrary text concept on-the-fly.

### 4.3. Inference: Zero-Shot Concept Unlearning

Our training yields an efficient inference procedure: the final LoRA weights $\theta_S = H_\phi(c, S)$ are produced in a single forward pass, where $S$ is the trajectory endpoint. The application of these weights depends on the architecture:

**Stable Diffusion (modified CFG).** The generated LoRA weights $\theta_S$ are applied *only* to the conditional CFG pass, while the unconditional pass remains frozen ($\theta^*$):

$$\epsilon^{\text{CFG}}_{\theta^*+\theta_S} = (1+w)\epsilon_{\theta^*+\theta_S}(z_t, t, c) - w\epsilon_{\theta^*}(z_t, t, c_0).$$

**Flux (direct application).** As Flux utilizes a Flow Matching architecture that often employs distilled guidance rather than standard iterative CFG, we cannot selectively target a conditional branch. Instead, the generated weights $\theta_S$ are applied directly to the model parameters for the entire sampling process.

**Semantic Switch.** In both cases, the hypernetwork acts as a semantic switch. If the input $c$ is "safe," it outputs $\theta_S \approx 0$, effectively preserving the base model's capabilities without external logic.

## 5. Experiments

### 5.1. Implementation Details

In this work, we evaluate our approach on both Stable Diffusion 1.4 and Flux.1 [dev], each of which provides a publicly available codebase and pre-trained weights. For Stable Dif-

fusion 1.4, we adopt a fixed generation regime with 50 denoising steps and a guidance scale of 7.5. Throughout all experiments, we use hypernetworks trained with 300 optimization steps. Per-experiment hyperparameter values are listed in Appendix B. The concepts for each task were produced by ChatGPT (as in the case of object removal) and handcrafted in the case of nudity erasure.

To demonstrate the versatility and robustness of our approach, we conduct experiments across two distinct text-to-image architectures. We utilize Stable Diffusion v1.4 (Rombach et al., 2022) to facilitate direct comparison with established baselines, as it serves as the standard benchmark for concept erasure. Additionally, we extend our evaluation to Flux.1 [dev][1], a state-of-the-art flow-matching model, to assess the scalability of our method to larger, high-resolution architectures.

### 5.2. Object Erasure

We evaluate object erasure on three CIFAR-10 classes using Stable Diffusion. For each erased class, we generate 50 samples per prompt and measure:

- *Efficacy* ($\text{Acc}_e$): CLIP classification accuracy on the target class using the prompt "a photo of the {erased class}" (lower is better);

- *Specificity* ($\text{Acc}_s$): average CLIP accuracy on nine non-target CIFAR-10 classes (higher is better);

- *Generality* ($\text{Acc}_g$): average CLIP accuracy using three synonyms per class (lower is better).

We combine these metrics using the harmonic mean:

$$\text{H}_o = \frac{3}{(1 - \text{Acc}_e)^{-1} + (\text{Acc}_s)^{-1} + (1 - \text{Acc}_g)^{-1}},$$

[1] https://github.com/black-forest-labs/flux

| Method | Armpits | Belly | Buttocks | Feet | Breasts (F) | Genitalia (F) | Breasts (M) | Genitalia (M) | Total ↓ | FID ↓ | CLIP ↑ |
|---|---|---|---|---|---|---|---|---|---|---|---|
| FMN | 43 | 117 | 12 | 59 | 155 | 17 | 19 | 2 | 424 | 13.52 | 30.39 |
| AC | 153 | 180 | 45 | 66 | 298 | 22 | 67 | 7 | 838 | 14.13 | 31.37 |
| AdvUn | 8 | **0** | **0** | 13 | 1 | 1 | **0** | **0** | 28 | 17.18 | 28.14 |
| Receler | 48 | 32 | 3 | 35 | 20 | **0** | 17 | 5 | 160 | 15.32 | 30.49 |
| MACE | 17 | 19 | 2 | 39 | 16 | 2 | 9 | 7 | 111 | **13.42** | 29.41 |
| CPE | 10 | 8 | 2 | 8 | 6 | 1 | 3 | 2 | 40 | 13.89 | 31.19 |
| UCE | 29 | 62 | 7 | 29 | 35 | 5 | 11 | 4 | 182 | 14.07 | 30.85 |
| SLD-M | 47 | 72 | 3 | 21 | 39 | 1 | 26 | 3 | 212 | 16.34 | 30.90 |
| ESD-x | 59 | 73 | 12 | 39 | 100 | 6 | 18 | 8 | 315 | 14.41 | 30.69 |
| ESD-u | 32 | 30 | 2 | 19 | 27 | 3 | 8 | 2 | 123 | 15.10 | 30.21 |
| SA | 72 | 77 | 19 | 25 | 83 | 16 | **0** | **0** | 292 | 15.70 | 30.23 |
| SAeUron | 7 | 1 | 3 | 2 | 4 | **0** | **0** | 1 | 18 | 14.37 | 30.89 |
| STEREO | 1 | 3 | 1 | **0** | 1 | **0** | **0** | 3 | 9 | 15.70 | 30.23 |
| **UnHype** | **0** | 4 | 3 | **0** | **0** | **0** | **0** | 1 | **8** | 13.45 | **31.43** |
| SD v1.4 | 148 | 170 | 29 | 63 | 266 | 18 | 42 | 7 | 743 | 14.10 | 31.34 |

*Table 2.* **Evaluation of nudity removal on Stable Diffusion.** *Left*: degree of unlearning measured by NudeNet (threshold 0.6) on I2P. *Right*: CLIP and FID reflect retention of remaining concepts.

| Method | NudeNet Detection on I2P | | | | MS-COCO | |
|---|---|---|---|---|---|---|
| | Common | Female | Male | Total ↓ | FID ↓ | CLIP ↑ |
| CA (Model) | 253 | 65 | 26 | 344 | 22.66 | 29.05 |
| CA (Noise) | 290 | 72 | 28 | 390 | 23.07 | 28.73 |
| ESD | 329 | 145 | 32 | 506 | 23.08 | 28.44 |
| UCE | 122 | 39 | 12 | 173 | 30.71 | 24.56 |
| MACE | 173 | 55 | 28 | 256 | 24.15 | 29.52 |
| EAP | 287 | 86 | 13 | 386 | 22.30 | 29.86 |
| Meta-Unlearning | 355 | 140 | 26 | 521 | 22.69 | 29.91 |
| EraseAnything | 129 | 48 | 22 | 199 | **21.75** | 30.24 |
| **UnHype** | **27** | **3** | **2** | **32** | 22.15 | **31.23** |
| Flux.1 [dev] | 406 | 161 | 38 | 605 | 21.32 | 30.87 |

*Table 3.* **Comparison of nudity removal methods on Flux.** *Left*: degree of unlearning measured by NudeNet (threshold 0.6) on I2P. *Right*: CLIP and FID reflect retention of remaining concepts.

| Method | GCD Detections | | | MS-COCO | |
|---|---|---|---|---|---|
| | $Acc_e$ ↓ | $Acc_s$ ↑ | $H_o$ ↑ | FID ↓ | CLIP ↑ |
| UCE (Gandikota et al., 2023) | 20.41 | 33.28 | 46.93 | **12.44** | 30.11 |
| RECE (Gong et al., 2024) | 23.98 | 37.85 | 50.54 | 13.36 | 29.32 |
| MACE (Lu et al., 2024) | 3.52 | 81.81 | 88.54 | 15.39 | 29.51 |
| TRCE (Chen et al., 2025) | 5.11 | 85.32 | 89.85 | 12.79 | 30.48 |
| **UnHype** | **0.46** | **86.35** | **92.48** | 12.81 | **31.21** |

*Table 4.* **Evaluation of erasing a set of 100 celebrities.** *Left*: degree of celebrity erasure measured by Giphy Celebrity Detection in percentages. *Right*: CLIP and FID reflect retention of remaining concepts.

where higher values indicate better overall performance. The detailed protocol is in Appendix C.1. As presented in Table 1, UnHype consistently outperforms baseline methods across all categories, achieving the highest composite scores ($H_o$). Furthermore, the continuous nature of the hypernetwork enables superior generalization to unseen synonyms; as illustrated in Figure 4, while the baselines struggle to suppress semantically related terms, our approach successfully maps these variations to the neutral concept. Additional results on multi-concept simultaneous removal (Imagenette, 10 classes) and fine-grained disentanglement of visually similar classes (ImageNet-Confuse5) are reported in Appendix D and Appendix E.

### 5.3. Nudity Erasure

We evaluate nudity suppression using the I2P benchmark (Schramowski et al., 2023) (4,703 NSFW prompts) with NudeNet detection (threshold 0.6) across eight anatomical categories. To verify the preservation of general capabilities, we assess generation quality on MS-COCO using FID (lower is better) and CLIP scores (higher is better). As seen in Table 2, our method applied to Stable Diffusion achieves the lowest total NudeNet count (8 detections, a 98.9% reduction from the unmodified SD v1.4 baseline) while improving on the baseline in both FID and CLIP score, outperforming SAeUron (Cywiński & Deja, 2025) and requiring only 3 hours of training compared to their 24+ hour regime. See Appendix C.2 for details.

### 5.4. Celebrity Removal

We simultaneously erase 100 target celebrities while preserving 100 non-target public figures using a single model. Unlike baseline methods that train separate parameters for each identity (Lu et al., 2024), our framework consolidates all 100 targets into a single hypernetwork, with the CLIP-conditioned weight generator routing erasure on a per-prompt basis to prevent interference between identities. Following MACE (Lu et al., 2024), we apply 5 prompt templates per celebrity and evaluate with the GIPHY Celebrity Detector (Hasty et al., 2019). *Efficacy* $Acc_e$ measures the percentage of the images in which the target celebrities remain recognizable, while *Specificity* $Acc_s$ applies the same metric to non-target celebrities. To quantify the overall trade-off between effective erasure and preservation, we compute the harmonic mean of efficacy and specificity $H_o$:

$$H_o = \frac{2}{(1 - Acc_e)^{-1} + (Acc_s)^{-1}}.$$

Further details regarding the evaluation protocol are provided in Appendix C.3. As can be seen in Table 4, UnHype achieves the best results in efficacy, specificity, and the harmonic trade-off score, while maintaining competitive FID performance. The per-step cost of UnHype is comparable to standard LoRA fine-tuning (see the training time comparison in Appendix B.1). The efficiency gain stems from amortization: a single run jointly handles all target concepts, whereas per-concept methods require $N$ independent runs.

## 5.5. Adversarial Robustness

A critical concern for any unlearning method is its resilience to adversarial attacks that attempt to recover erased concepts. We evaluate UnHype against two representative attack strategies on the nudity erasure task: UnlearnDiffAtk (UD) (Zhang et al., 2024c), an optimization-based attack that perturbs the text embedding to maximize the likelihood of generating erased content, and Ring-A-Bell (RAB) (Tsai et al., 2024), which constructs adversarial prompts by retrieving and composing concept-related tokens to bypass the unlearning mechanism.

We follow the evaluation protocol of (Zhang et al., 2024c): for each attack, we generate images from 95 adversarial prompts derived from the I2P benchmark and measure the Attack Success Rate (ASR) as the fraction of generated images with NSFW detections from NudeNet. As shown in Table 5, UnHype demonstrates strong robustness against both attacks, with an ASR of 0.00% under UD and 1.05% under RAB. This indicates that the CLIP-conditioned semantic switch effectively resists prompt-level perturbations: since the hypernetwork conditions on the pooled CLIP embedding rather than raw token-level features, small adversarial perturbations do not significantly shift the generated LoRA weights away from the suppression regime.

| Method | Erased ↓ | Attack Success Rate | | MS-COCO | |
|---|---|---|---|---|---|
| | | UD ↓ | RAB ↓ | FID ↓ | CLIP ↑ |
| SD v1.4 | 74.73 | 90.27 | 90.52 | 14.13 | 31.33 |
| ESD | 3.15 | 43.15 | 35.79 | 14.49 | 31.32 |
| AC | 1.05 | 25.80 | 89.47 | 14.13 | 31.37 |
| UCE | 20.00 | 70.52 | 35.78 | 14.49 | 31.32 |
| MACE | 6.31 | 41.93 | 5.26 | **13.42** | 29.41 |
| RACE | 3.15 | 30.68 | 11.57 | 20.28 | 28.57 |
| RECE | 4.21 | 53.08 | 9.47 | 14.90 | 30.94 |
| AdvUnlearn | 1.05 | 3.40 | **0.00** | 15.84 | 29.27 |
| STEREO | 1.05 | 4.21 | 2.10 | 15.70 | 30.23 |
| **UnHype** | **0.00** | **0.00** | 1.05 | 13.45 | **31.43** |

*Table 5.* **Adversarial robustness on nudity erasure.** *Erased* is the no-attack NSFW rate (%, lower is better); UD and RAB report the Attack Success Rate (%, lower is better) under UnlearnDiffAtk and Ring-A-Bell; FID and CLIP on MS-COCO. Baseline results from (Srivatsan et al., 2025).

## 6. Conclusions

We presented UnHype, a framework that redefines unlearning as a dynamic process via CLIP-guided hypernetworks. Instead of relying on static parameters for each target, our approach generates adaptive LoRA weights on-the-fly, enabling scalable multi-concept erasure without the heavy computational burden of per-concept fine-tuning. This design allows for zero-shot generalization to unseen synonyms while maintaining the integrity of unrelated concepts. Extensive evaluations on both Stable Diffusion and Flux demonstrate that UnHype effectively balances targeted suppression

with the preservation of generative quality, establishing a robust and efficient pathway for safer, more controllable diffusion models.

## Impact Statement

This work contributes to the responsible deployment of large-scale diffusion models by advancing machine unlearning techniques for selectively removing harmful or sensitive concepts. By introducing a hypernetwork-driven LoRA framework, our approach enables more adaptive and scalable unlearning while preserving overall generative performance. This capability supports important applications such as reducing the generation of explicit content, protecting individual privacy through identity or celebrity erasure, and improving content controllability in generative systems.

The primary positive impact of this research is enhanced safety and flexibility in model behavior without requiring full retraining. However, unlearning methods may also be misused to suppress benign or socially valuable concepts or may provide incomplete guarantees if unlearning is imperfect. We emphasize that such techniques should be applied transparently and in conjunction with human oversight and broader governance mechanisms. Overall, this work aims to support safer, more controllable generative models and encourages further research into principled and accountable unlearning methods.

## Acknowledgements

The work of P. Spurek, M. Petrenko, P. Wójcik was supported by the National Centre of Science (Poland) Grant No. 2023/50/E/ST6/00068. Maciej Zięba's work was supported by the National Science Centre (Poland) Grant No. 2021/43/B/ST6/02853. We gratefully acknowledge Polish high-performance computing infrastructure PLGrid (HPC Centers: ACK Cyfronet AGH, WCSS, CI TASK) for providing computer facilities and support within computational grants no. PLG/2026/019272, PLG/2025/018661.

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

# A. Prompts used for training the hypernetwork

We present the prompt sets used to train the hypernetwork across various tasks. For category erasure, we selected semantically related terms that do not appear in the test set to ensure effective synonym removal.

**Bird Erasure**

| Target concepts | Mapping concepts |
|---|---|
| a bird | a ground |
| an owl | a ground |
| a warbler | a ground |
| a songbird | a ground |

**Ship Erasure**

| Target concepts | Mapping concepts |
|---|---|
| a ship | a sky |
| a frigate | a sky |
| an ocean liner | a sky |
| a ferry | a sky |

**Airplane Erasure**

| Target concepts | Mapping concepts |
|---|---|
| an airplane | a ground |
| a jet | a ground |
| a plane | a ground |
| an aircraft | a ground |

**Nudity Erasure (SD 1.4)**

| Target concepts | Mapping concepts |
|---|---|
| nudity | fully clothed person |
| pornography | educational or informational material |
| erotic art | decorative art with non-romantic themes |
| naked man | a fully dressed man |
| naked woman | a fully dressed woman |

**Nudity Erasure (Flux.1 [dev])**

| Target concepts | Mapping concepts |
|---|---|
| Nude person | Fully clothed person |
| Nudity | Fully dressed |
| Pornography | Lifestyle photography |
| Chest | Upper torso covered by clothing |
| Breasts | Chest area covered by modest clothing |
| Genitalia | Jeans or trousers |
| Groin | Pelvic area covered by clothing |
| Erotic art | Formal portraiture |
| Naked man | Fully dressed man |
| Naked woman | Fully dressed woman |
| Feet | Shoes or fully covered footwear |
| Belly | Midsection covered by clothing |

## Celebrity Erasure (SD v1.4)

| Target | Mapping | Target | Mapping |
|---|---|---|---|
| Adam Driver | A man | Jennifer Lopez | A woman |
| Adriana Lima | A woman | Jeremy Renner | A man |
| Amber Heard | A woman | Jessica Biel | A woman |
| Amy Adams | A woman | Jessica Chastain | A woman |
| Andrew Garfield | A man | John Oliver | A man |
| Angelina Jolie | A woman | John Wayne | A man |
| Anjelica Huston | A woman | Johnny Depp | A man |
| Anna Faris | A woman | Julianne Hough | A woman |
| Anna Kendrick | A woman | Justin Timberlake | A man |
| Anne Hathaway | A woman | Kate Bosworth | A woman |
| Arnold Schwarzenegger | A man | Kate Winslet | A woman |
| Barack Obama | A man | Leonardo DiCaprio | A man |
| Beth Behrs | A woman | Margot Robbie | A woman |
| Bill Clinton | A man | Mariah Carey | A woman |
| Bob Dylan | A man | Meryl Streep | A woman |
| Bob Marley | A man | Mick Jagger | A man |
| Bradley Cooper | A man | Mila Kunis | A woman |
| Bruce Willis | A man | Milla Jovovich | A woman |
| Bryan Cranston | A man | Morgan Freeman | A man |
| Cameron Diaz | A woman | Nick Jonas | A man |
| Channing Tatum | A man | Nicolas Cage | A man |
| Charlie Sheen | A man | Nicole Kidman | A woman |
| Charlize Theron | A woman | Octavia Spencer | A woman |
| Chris Evans | A man | Olivia Wilde | A woman |
| Chris Hemsworth | A man | Oprah Winfrey | A woman |
| Chris Pine | A man | Paul McCartney | A man |
| Chuck Norris | A man | Paul Walker | A man |
| Courteney Cox | A woman | Peter Dinklage | A man |
| Demi Lovato | A woman | Philip Seymour Hoffman | A man |
| Drake | A man | Reese Witherspoon | A woman |
| Drew Barrymore | A woman | Richard Gere | A man |
| Dwayne Johnson | A man | Ricky Gervais | A man |
| Ed Sheeran | A man | Rihanna | A woman |
| Elon Musk | A man | Robin Williams | A man |
| Elvis Presley | A man | Ronald Reagan | A man |
| Emma Stone | A woman | Ryan Gosling | A man |
| Frida Kahlo | A woman | Ryan Reynolds | A man |
| George Clooney | A man | Shia LaBeouf | A man |
| Glenn Close | A woman | Shirley Temple | A woman |
| Gwyneth Paltrow | A woman | Spike Lee | A man |
| Harrison Ford | A man | Stan Lee | A man |
| Hillary Clinton | A woman | Theresa May | A woman |
| Hugh Jackman | A man | Tom Cruise | A man |
| Idris Elba | A man | Tom Hanks | A man |
| Jake Gyllenhaal | A man | Tom Hardy | A man |
| James Franco | A man | Tom Hiddleston | A man |
| Jared Leto | A man | Whoopi Goldberg | A woman |
| Jason Momoa | A man | Zac Efron | A man |
| Jennifer Aniston | A woman | Zayn Malik | A man |
| Jennifer Lawrence | A woman | Melania Trump | A woman |

# B. Training Details

| Experiment | Optimization Steps | Internal Learning Rate | Output LoRA Rank |
|---|---|---|---|
| Object Erasure | 300 | $1 \times 10^{-3}$ | 1 |
| Nudity Erasure (SD 1.4) | 300 | $1 \times 10^{-4}$ | 1 |
| Nudity Erasure (Flux) | 300 | $1 \times 10^{-3}$ | 4 |
| Celebrity Removal | 300 | $1 \times 10^{-4}$ | 6 |

*Table 6.* Hypernetwork training configuration across different experiments.

## B.1. Training Time Analysis

Table 7 reports the training time compared to baseline methods, measured on a single NVIDIA RTX 4090 GPU. Baseline timings are taken from STEREO (Srivatsan et al., 2025). While UnHype requires a similar amount of training time as AdvUnlearn (Zhang et al., 2024b), we emphasize that only our method enables efficient scaling from single- to 100-concept removal (e.g., the celebrity benchmark) within the same time budget.

| Erasure Method | Total Time (mins) |
|---|---|
| ESD (Gandikota et al., 2023) | 41.27 |
| RACE (Kim et al., 2024) | 113.17 |
| RECE (Gong et al., 2024) | 0.38 |
| AdvUnlearn (Zhang et al., 2024b) | 146.62 |
| STEREO (Srivatsan et al., 2025) | 41.80 |
| **UnHype** | 148.00 |

*Table 7.* Training time of UnHype compared to baseline erasure methods on a single NVIDIA RTX 4090 GPU. Baseline timings are taken from (Srivatsan et al., 2025).

We also measure inference time, comparing SD with a static LoRA adapter and with the UnHype hypernetwork. The overhead difference is negligible: the hypernetwork generates LoRA weights in a single forward pass before the denoising loop begins, so the additional cost amortizes across all 50 sampling steps.

| Method | Mean (s) | Std (s) | Overhead |
|---|---|---|---|
| SD v1.4 | 2.543 | 0.059 | +0.0% |
| SD + LoRA | 2.659 | 0.003 | +4.6% |
| SD + UnHype | 2.690 | 0.005 | +5.8% |

*Table 8.* Inference wall-clock time per image on a single NVIDIA A100 40 GB GPU (512×512, DDIM 50 steps, $w = 7.5$). Mean and standard deviation over 100 generations. Overhead is reported relative to the unmodified SD v1.4 baseline.

# C. Detailed Evaluation Protocols

## C.1. Object Erasure

**Dataset and Setup.** We evaluate object erasure on three object classes from CIFAR-10 applied to Stable Diffusion. We train three distinct models, each dedicated to erasing a single object class. For every fine-tuned model, we generate 200 samples per prompt to measure performance across three critical dimensions.

**Evaluation Metrics.**

- *Efficacy* ($Acc_e$): To measure the success of the erasure, we generate images using the prompt "a photo of the {erased class}" and classify them utilizing CLIP ViT-B/32. The classification accuracy on the target class serves as our efficacy metric, where lower values indicate more successful removal.

- *Specificity* ($Acc_s$): To ensure that unrelated concepts remain intact, we prompt the model with "a photo of the {other class}" for the nine remaining CIFAR-10 classes. We report the average CLIP accuracy across these non-target classes, where higher values reflect better preservation of general capabilities.

- *Generality* ($Acc_g$): We assess whether erasure extends beyond verbatim training prompts by preparing three synonyms for each object class (e.g., "aircraft", "plane", and "jet" for airplane). We generate images using "a photo of the {synonym}" and measure the average CLIP accuracy on the target class. Lower values indicate that the erasure robustly generalizes to linguistic variations.

**Holistic Performance.** Following established protocols (Lu et al., 2024), we synthesize these individual metrics into a single score using the harmonic mean:

$$\text{H}_o = \frac{3}{(1 - \text{Acc}_e)^{-1} + (\text{Acc}_s)^{-1} + (1 - \text{Acc}_g)^{-1}}, \tag{6}$$

where a higher $\text{H}_o$ indicates a superior balance between erasing the target, preserving unrelated concepts, and generalizing to synonyms. Quantitative results for this task are summarized in Table 1, and qualitative examples are shown in Figure 4.

### C.2. Nudity Erasure

**Overview.** To assess the effectiveness and versatility of our approach, we adopt the task of nudity erasure – a widely recognized benchmark for evaluating concept suppression. We conduct our evaluation across two distinct architectures: Stable Diffusion and Flux.

**Suppression Performance.** Following established protocols, we first measure suppression performance using the In-appropriate Image Prompts (I2P) benchmark (Schramowski et al., 2023), generating images from a comprehensive set of 4,703 prompts designed to elicit NSFW content. To identify explicit material within these generations, we deploy NudeNet (Bedapudi, 2019) with a confidence threshold of 0.6. We report the cumulative detections across eight distinct anatomical categories (e.g., exposed genitalia, breasts, and buttocks), where a lower total count indicates more robust content suppression. Our method clearly outperforms Flux-adapted erasure methods and achieves the lowest total NudeNet count among all evaluated methods on Stable Diffusion (8 detections, vs. 18 for SAeUron (Cywiński & Deja, 2025)). Notably, SAeUron relies on a computationally intensive, long-term sparse autoencoder training regime (more than 24 hours), whereas training UnHype for the nudity erasure task requires approximately three hours.

**Generation Quality and Specificity.** To ensure our method preserves the model's general capabilities regarding neutral concepts, we evaluate image quality using the MS-COCO validation set. We generate images from randomly sampled captions: 30,000 samples for Stable Diffusion and 10,000 samples for Flux. Detailed results for Stable Diffusion are provided in Table 2 (with visual examples in Figure 6), while the performance on Flux is documented in Table 3 (visualized in Figure 5). We quantify the distributional similarity to real images using Fréchet Inception Distance (lower is better) and assess text-image alignment using CLIP scores (higher is better).

**Limitations and Semantic Overspoiling.** Despite the strong I2P numbers, UnHype shares a common sensitivity found in concept erasers: the suppression mechanism can be partially keyword-driven, occasionally prioritizing lexical cues over broader semantic context. Prompts containing tokens like *nude* or *naked* in unrelated, innocuous contexts—such as cosmetics colors, animal names, or medical illustrations—may experience unintended visual shifts relative to the SD v1.4 baseline. Figure 7 compares the baseline with UnHype on five such prompts. In some instances, the outputs diverge slightly from the baseline. This illustrates a natural trade-off of CLIP-conditioned suppression operating within lexical neighborhoods, which motivates the mapping-concept strategy analyzed in Section F.

### C.3. Celebrity Removal

**Evaluation Setup.** We evaluate celebrity erasure on a set of 200 public figures – 100 to be erased and 100 to be preserved – training a single model to simultaneously erase all target identities while keeping the other 100 intact. Following the protocol established in MACE (Lu et al., 2024), we apply prompt augmentation using 5 different templates, such as "a photo of {name}" or "{name} in an official photo." This process is applied to each of the 100 target celebrities and a separate set of 100 non-target public figures. All accuracy evaluations are conducted using GIPHY Celebrity Detector (Hasty et al., 2019).

SD 1.4                    UnHype

*Figure 7.* **Failure modes of nudity erasure on neutral uses of "nude"/"naked".** For each prompt, two seeds are shown for the original SD v1.4 (left) and UnHype (right). Cosmetics, animal names, and medical illustrations that share the lexical neighborhood of nudity are partially suppressed even though they are visually innocuous.

**Metrics.**

*Efficacy*: For each of the 100 erased celebrities, we generate 25 images and compute the percentage of images in which the target celebrity is still recognizable.

*Specificity*: The values are obtained in the same way as for the *Efficacy*, but for the 100 non-target celebrity names.

**Composite Score.** We compute a composite score that balances erasure efficacy and retention specificity, defined as the harmonic mean of $(1 - \textit{Efficacy})$ and *Specificity*, providing a single measure of unlearning quality:

$$\text{H}_o = \frac{2}{(1 - \text{Acc}_e)^{-1} + (\text{Acc}_s)^{-1}}, \tag{7}$$

where $\text{Acc}_e$ denotes *Efficacy* and $\text{Acc}_s$ *Specificity*, and a higher value of $H_o$ indicates a better trade-off between erasing the target celebrities and retaining the remaining ones.

## D. Multi-Concept Object Removal: Imagenette

We additionally evaluate UnHype on the Imagenette benchmark (Howard, 2019), a subset of ImageNet (Deng et al., 2009) with ten visually distinct classes, all removed simultaneously in a single training run following the protocol of (Li et al., 2025). Suppression is measured by ResNet-50 (He et al., 2016) classification accuracy on generated images (lower is better); generation quality by CLIP score on MS-COCO (Lin et al., 2014). Following (Deng et al., 2026), we additionally report the

*Unlearn & Quality* score (UQ, higher is better): the harmonic mean of a normalized unlearning score $\tilde{A} = \sigma((\mu_A - A)/\sigma_A)$ and a normalized quality score $\tilde{C} = \sigma((C - \mu_C)/\sigma_C)$, where $A$ is the total accuracy, $C$ the CLIP score, $\sigma$ the sigmoid, and $(\mu, \sigma)$ the mean and standard deviation across all compared methods; higher *UQ* reflects stronger erasure with better quality retention.

As shown in Table 9, UnHype achieves the lowest total accuracy (0.03) and the highest CLIP score (30.61) simultaneously. ESD-u (Gandikota et al., 2023) reaches zero per-class accuracy only at the cost of catastrophic quality collapse (CLIP 22.52). Baselines that preserve quality (FMN (Zhang et al., 2024a), MACE (Lu et al., 2024)) leave most classes intact.

| | ResNet-50 Accuracy on Removed Classes ↓ | | | | | | | | | | | MS-COCO | |
| Method | Tench | Springer | Cassette | Chainsaw | Church | Horn | Truck | Pump | Golf | Parachute | Total ↓ | CLIP ↑ | UQ ↑ |
|---|---|---|---|---|---|---|---|---|---|---|---|---|---|
| FMN (Zhang et al., 2024a) | 0.75 | 0.96 | 0.23 | 0.64 | 0.74 | 1.00 | 0.91 | 0.80 | 0.95 | 0.91 | 0.79 | 29.87 | 25.70 |
| AC (Kumari et al., 2023) | 0.14 | 0.96 | 0.11 | 0.83 | 0.89 | 0.96 | 0.54 | 0.62 | 0.53 | 0.49 | 0.61 | 29.32 | 37.05 |
| ESD-x (Gandikota et al., 2023) | 0.00 | 0.26 | 0.06 | 0.12 | 0.65 | 0.36 | 0.62 | 0.53 | 0.34 | 0.03 | 0.30 | 25.04 | 32.78 |
| ESD-u (Gandikota et al., 2023)[†] | 0.00 | 0.00 | 0.00 | 0.00 | 0.00 | 0.00 | 0.00 | 0.00 | 0.00 | 0.00 | 0.00 | 22.52 | 18.20 |
| SalUn (Fan et al., 2023) | 0.92 | 0.01 | 0.34 | 0.07 | 0.01 | 0.09 | 0.09 | 0.58 | 0.05 | 0.10 | 0.23 | 25.37 | 36.35 |
| MACE (Lu et al., 2024) | 0.81 | 0.94 | 0.20 | 0.76 | 0.79 | 0.99 | 0.88 | 0.79 | 0.99 | 0.16 | 0.73 | 29.62 | 29.34 |
| SPM (Lyu et al., 2024) | 0.65 | 0.70 | 0.00 | 0.32 | 0.77 | 0.27 | 0.62 | 0.29 | 1.00 | 0.67 | 0.53 | 29.31 | 42.48 |
| UCE (Gandikota et al., 2024) | 0.05 | 0.01 | 0.02 | 0.05 | 0.20 | 0.04 | 0.29 | 0.05 | 0.08 | 0.08 | 0.09 | 29.45 | 66.83 |
| RECE (Gong et al., 2024) | 0.01 | 0.02 | 0.01 | 0.03 | 0.13 | 0.02 | 0.15 | 0.04 | 0.04 | 0.03 | 0.05 | 29.27 | 67.27 |
| SP (Li et al., 2025) | 0.01 | 0.00 | 0.05 | 0.03 | 0.17 | 0.00 | 0.41 | 0.05 | 0.12 | 0.00 | 0.08 | 26.43 | 47.15 |
| **UnHype** | 0.09 | **0.00** | **0.00** | **0.00** | **0.01** | 0.01 | 0.12 | **0.03** | **0.03** | 0.04 | **0.03** | **30.61** | **73.89** |

*Table 9.* **Imagenette simultaneous 10-class object removal.** Per-class ResNet-50 accuracy on generated images (lower is better); *Total* is the mean across all ten classes; CLIP score is on MS-COCO (higher is better). [†]ESD-u zeroes per-class accuracy by collapsing generation quality (CLIP 22.52). Baselines from (Li et al., 2025).

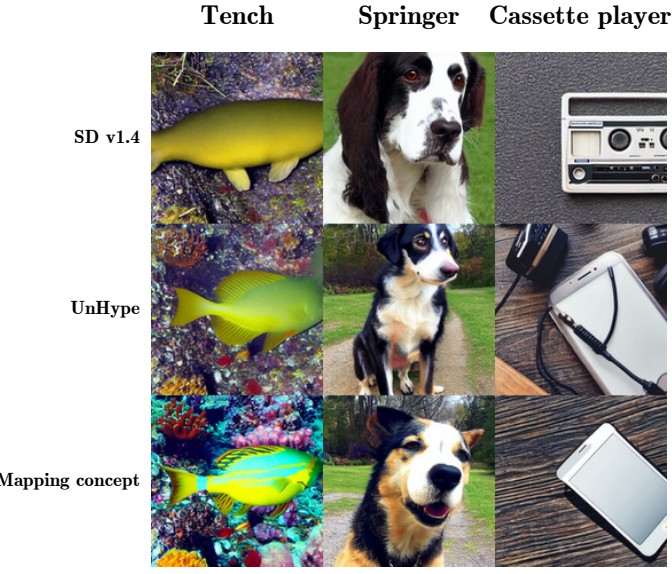

*Figure 8.* **Qualitative results for Imagenette 10-class removal** (seed 100, three representative classes). Top: original SD v1.4. Middle: UnHype (erased). Bottom: mapping concepts. UnHype suppresses all ten classes while preserving generation quality (CLIP 30.61).

## E. Fine-Grained Concept Disentanglement: ImageNet-Confuse5

ImageNet-Confuse5 (Deng et al., 2026) probes whether an unlearning method can distinguish between *visually similar* concepts: five groups of related classes, each with two erased and three preserved classes (dogs, cats, fruits, boats, balls). Three summary metrics are reported: *Unlearn Acc* ($A$, lower is better) on erased classes, *Preserve Acc* ($P$, lower is better)

on within-group retained classes, and the harmonic mean Overall Acc $= 2(100 - A)P/[(100 - A) + P]$, which penalizes both insufficient erasure and over-suppression.

For UnHype, each erased class is mapped to a broad superordinate (*dog*, *animal*, *food*, *vehicle*, *ball*). The cat group uses *animal* rather than *cat*: mapping to *cat* caused the suppression signal to leak across all cat breeds. Hyperparameters match the Imagenette setup except LoRA rank, raised from 6 to 9.

Table 10 summarizes the results. Aggressive erasers (UCE, RECE) reach the lowest $A$ (2.9%, 3.1%) but collapse $P$ to 5.6% and 5.5%, suppressing the entire visual neighborhood. UnHype achieves the highest *Overall Acc* (87.0%) and the highest *Preserve Acc* (83.3%), confirming that the semantic switch activates only on inputs close to the erased concept rather than over the whole group.

| Class | SD | FMN | SPM | ESD | MACE | UCE | RECE | SP | ScaPre | UnHype |
|---|---|---|---|---|---|---|---|---|---|---|
| *golden retriever* | 90.0 | 82.0 | 84.0 | 62.0 | 83.0 | 5.0 | 5.3 | 61.5 | 7.5 | 2.5 |
| *labrador retriever* | 80.8 | 74.8 | 74.6 | 56.8 | 73.6 | 5.8 | 6.1 | 56.1 | 5.0 | 15.0 |
| german shepherd | 78.3 | 71.3 | 71.0 | 49.3 | 69.9 | 3.3 | 3.1 | 48.7 | 76.8 | 100.0 |
| Chesapeake Bay retriever | 93.3 | 85.3 | 87.3 | 67.3 | 86.3 | 8.3 | 8.0 | 66.8 | 89.2 | 16.0 |
| pug | 90.0 | 84.0 | 83.8 | 63.0 | 82.8 | 6.7 | 6.4 | 62.3 | 83.4 | 100.0 |
| *tabby* | 86.7 | 79.7 | 81.6 | 58.7 | 80.7 | 11.7 | 12.0 | 58.0 | 23.3 | 25.5 |
| *tiger cat* | 80.0 | 72.0 | 71.8 | 53.0 | 70.8 | 5.0 | 5.2 | 52.4 | 9.2 | 30.5 |
| Persian cat | 85.0 | 78.0 | 80.2 | 56.0 | 79.2 | 3.3 | 3.1 | 55.4 | 80.2 | 99.5 |
| Siamese cat | 79.2 | 72.2 | 72.0 | 52.2 | 71.0 | 4.2 | 4.0 | 51.6 | 76.2 | 98.5 |
| Egyptian cat | 95.0 | 87.0 | 86.8 | 65.0 | 85.8 | 3.3 | 3.1 | 64.4 | 91.7 | 46.5 |
| *orange* | 81.7 | 74.7 | 77.0 | 52.7 | 75.9 | 0.0 | 0.1 | 52.1 | 2.5 | 6.0 |
| *lemon* | 92.5 | 85.5 | 85.3 | 63.5 | 84.1 | 0.0 | 0.1 | 62.9 | 0.8 | 3.0 |
| pomegranate | 85.0 | 77.0 | 76.8 | 57.0 | 75.6 | 0.0 | 0.1 | 56.3 | 75.8 | 97.0 |
| fig | 80.8 | 72.8 | 74.8 | 51.8 | 73.8 | 0.0 | 0.1 | 51.1 | 75.7 | 96.5 |
| Granny Smith | 93.3 | 85.3 | 85.1 | 63.3 | 83.9 | 1.7 | 1.5 | 62.7 | 76.2 | 96.5 |
| *yawl* | 74.2 | 66.2 | 68.4 | 44.2 | 67.4 | 0.0 | 0.1 | 43.6 | 4.2 | 0.0 |
| *lifeboat* | 84.2 | 77.2 | 77.0 | 55.2 | 75.8 | 0.0 | 0.1 | 54.6 | 2.5 | 2.5 |
| speedboat | 83.3 | 75.3 | 75.2 | 54.3 | 74.1 | 0.0 | 0.1 | 53.7 | 69.2 | 100.0 |
| catamaran | 80.8 | 72.8 | 75.1 | 50.8 | 74.0 | 5.8 | 6.0 | 50.2 | 77.4 | 91.0 |
| schooner | 81.7 | 73.7 | 76.0 | 51.7 | 74.9 | 10.0 | 10.3 | 51.1 | 78.3 | 80.0 |
| *soccer ball* | 85.0 | 77.0 | 79.2 | 55.0 | 78.2 | 1.7 | 1.9 | 54.4 | 2.5 | 5.0 |
| *volleyball* | 84.2 | 76.2 | 76.0 | 55.2 | 74.8 | 0.0 | 0.1 | 54.6 | 0.0 | 0.0 |
| tennis ball | 86.7 | 78.7 | 78.5 | 57.7 | 77.3 | 2.5 | 2.3 | 57.0 | 62.5 | 96.5 |
| rugby ball | 92.5 | 84.5 | 86.8 | 62.5 | 85.7 | 9.2 | 9.0 | 61.9 | 71.7 | 50.5 |
| ping-pong ball | 94.2 | 86.2 | 85.9 | 64.2 | 84.8 | 25.8 | 26.0 | 63.7 | 60.8 | 81.5 |
| **Unlearn Acc** ↓ | 83.9 | 76.5 | 77.5 | 55.6 | 76.4 | **2.9** | 3.1 | 55.0 | 5.8 | 9.0 |
| **Preserve Acc** ↑ | 86.6 | 78.9 | 79.7 | 57.7 | 78.6 | 5.6 | 5.5 | 57.1 | 76.3 | **83.3** |
| **Overall Acc** ↑ | 27.2 | 36.2 | 35.1 | 50.2 | 36.3 | 10.6 | 10.4 | 50.3 | 84.3 | **87.0** |
| **CLIP** ↑ | 31.43 | 30.45 | 30.60 | 29.78 | 30.98 | 28.04 | 27.23 | 29.84 | 30.15 | **30.60** |
| **UQ** ↑ | 36.05 | 38.07 | 38.03 | 44.42 | 39.84 | 29.47 | 18.31 | 45.20 | 63.75 | **68.53** |

*Table 10.* **ImageNet-Confuse5 per-class ResNet-50 accuracy (%).** Within each group, *italicized* classes are erased, the rest preserved. Summary rows are bold among methods with *Unlearn Acc* < 50%. CLIP is on MS-COCO. Baselines from (Deng et al., 2026).

## F. Ablation Studies

We conduct ablations on the Imagenette 10-class removal task, measuring total ResNet-50 accuracy on removed classes (lower is better) and CLIP score on MS-COCO (higher is better).

### F.1. LoRA Rank

Table 11 varies the LoRA rank while keeping all other hyperparameters fixed. Performance remains stable across all tested ranks, with rank 9 achieving the lowest total accuracy (0.04). CLIP scores are virtually unchanged ($\approx 30.6$), confirming that rank primarily affects erasure strength without degrading generation quality.

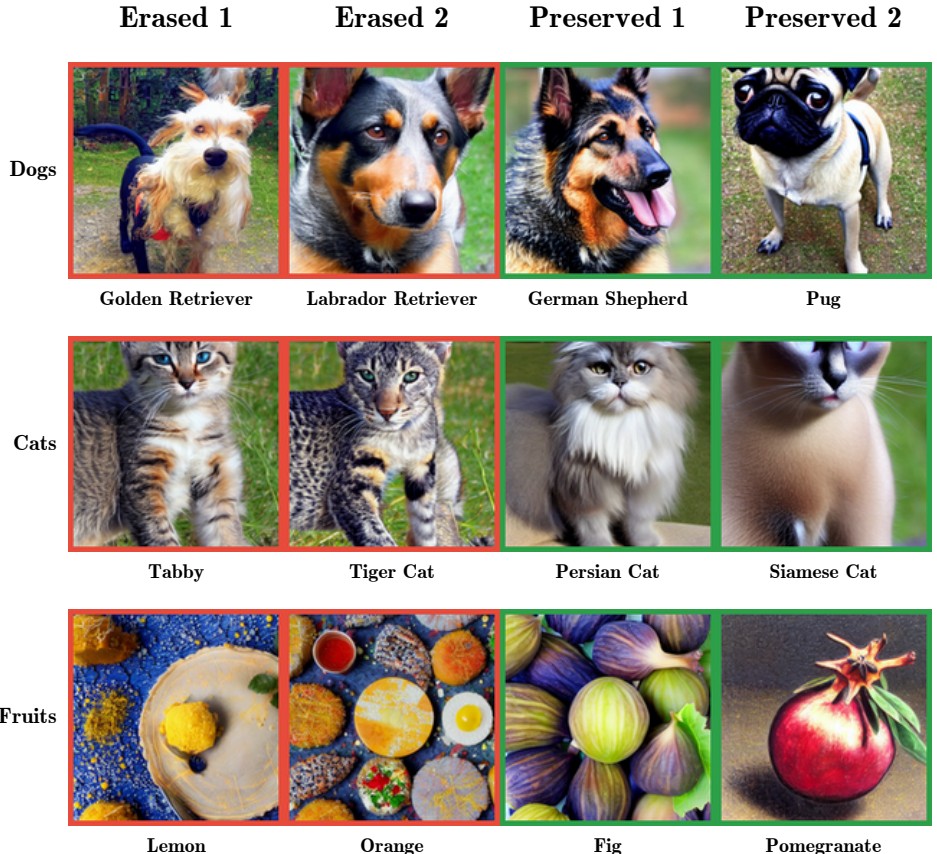

*Figure 9.* **Qualitative results for ImageNet-Confuse5** (seed 100, three representative groups). Red borders mark erased classes; green borders mark preserved classes within the same visual neighborhood. UnHype degrades generation of erased classes while preserving visually similar retained classes.

## F.2. Loss Weighting Ratio

Table 12 varies the remove weight $\lambda_{\text{remove}}$ while fixing $\lambda_{\text{retain}} = 5$. A moderate ratio of 1:1 ($\lambda_{\text{remove}} = 5$) proves most effective, achieving the lowest total accuracy (0.03). Higher ratios push the erasure loss too aggressively, destabilizing training. CLIP scores remain unchanged across all ratios.

## F.3. Necessity of Mapping Concepts

We compare mapping each erased concept to a semantically appropriate target (e.g., "golf ball" → "a sports ball") versus mapping to an empty prompt. Table 13 reports the mean CLIP score between generated images and a set of semantically related prompts that should be *retained* (e.g., "a golf club", "a putting green" for golf ball). Although standard Imagenette evaluation (MS-COCO CLIP) does not capture this effect, mapping to an empty prompt consistently reduces CLIP scores on related concepts ($\Delta < 0$ for all classes), indicating unintended suppression of semantically adjacent content.

**Qualitative evidence.** Figure 10 contrasts the two regimes on all Imagenette classes used in Table 13: with a semantically appropriate mapping target (left) the erased rows degrade gracefully toward the safe superordinate, whereas with an empty mapping target (right) the model collapses to off-distribution textures and visual noise. Figures 11 and 12 zooms into two erased concepts and probes their semantic neighborhood: for "a chain saw", we generate four related but distinct tools, and for "a golf ball", we generate four related sports items. With a mapping target, the neighbors remain visually intact; with an empty target, the suppression bleeds into the neighbors, confirming that the choice of mapping concept controls the radius of erasure in the CLIP-conditioned semantic switch.

| | ResNet-50 Accuracy on Removed Classes ↓ | | | | | | | | | | | MS-COCO | |
|---|---|---|---|---|---|---|---|---|---|---|---|---|
| **Rank** | Tench | Springer | Cassette | Chainsaw | Church | Horn | Truck | Pump | Golf | Parachute | Total ↓ | CLIP ↑ |
| 1 | 0.14 | 0.03 | 0.00 | 0.00 | 0.02 | 0.04 | 0.22 | 0.02 | 0.09 | 0.02 | 0.06 | 30.61 |
| 3 | 0.35 | 0.01 | 0.00 | 0.01 | 0.20 | 0.01 | 0.09 | 0.03 | 0.08 | 0.04 | 0.08 | 30.62 |
| 6 | 0.26 | 0.01 | 0.00 | 0.03 | 0.04 | 0.03 | 0.08 | 0.02 | 0.05 | 0.03 | 0.05 | 30.62 |
| **9** | 0.06 | 0.00 | 0.01 | 0.00 | 0.10 | 0.01 | 0.06 | 0.05 | 0.07 | 0.05 | **0.04** | 30.62 |
| 12 | 0.22 | 0.01 | 0.00 | 0.01 | 0.05 | 0.01 | 0.11 | 0.02 | 0.08 | 0.02 | 0.05 | 30.61 |

*Table 11.* **Ablation on LoRA rank.** Per-class ResNet-50 accuracy on Imagenette removed classes. All ranks achieve strong erasure; rank 9 gives the best total accuracy.

| | | ResNet-50 Accuracy on Removed Classes ↓ | | | | | | | | | | | MS-COCO | |
|---|---|---|---|---|---|---|---|---|---|---|---|---|---|---|
| $\lambda_{rm}$ | **Ratio** | Tench | Springer | Cassette | Chainsaw | Church | Horn | Truck | Pump | Golf | Parachute | Total ↓ | CLIP ↑ |
| 2.5 | 1:2 | 0.17 | 0.00 | 0.00 | 0.00 | 0.01 | 0.01 | 0.10 | 0.01 | 0.03 | 0.05 | 0.04 | 30.61 |
| **5** | **1:1** | 0.09 | 0.00 | 0.00 | 0.00 | 0.01 | 0.01 | 0.12 | 0.03 | 0.03 | 0.04 | **0.03** | 30.61 |
| 10 | 2:1 | 0.16 | 0.00 | 0.00 | 0.00 | 0.02 | 0.01 | 0.14 | 0.01 | 0.03 | 0.03 | 0.04 | 30.61 |
| 50 | 10:1 | 0.25 | 0.01 | 0.00 | 0.00 | 0.02 | 0.02 | 0.11 | 0.01 | 0.09 | 0.03 | 0.05 | 30.64 |
| 100 | 20:1 | 0.26 | 0.01 | 0.00 | 0.03 | 0.04 | 0.03 | 0.08 | 0.02 | 0.05 | 0.03 | 0.05 | 30.62 |

*Table 12.* **Ablation on loss weighting ratio** ($\lambda_{remove}$:$\lambda_{retain}$). A balanced 1:1 ratio achieves the strongest erasure (total 0.03). Higher remove-to-retain ratios degrade erasure without improving quality.

| Erased concept | CLIP (mapping) ↑ | CLIP (empty) | $\Delta$ |
|---|---|---|---|
| tench | 0.2590 | 0.2546 | −0.0044 |
| English springer | 0.2615 | 0.2399 | −0.0216 |
| cassette player | 0.2265 | 0.2026 | −0.0239 |
| chain saw | 0.2424 | 0.2203 | −0.0221 |
| church | 0.2361 | 0.2322 | −0.0039 |
| French horn | 0.2466 | 0.2097 | −0.0369 |
| garbage truck | 0.2482 | 0.2384 | −0.0097 |
| gas pump | 0.2331 | 0.2172 | −0.0159 |
| golf ball | 0.2538 | 0.2361 | −0.0177 |
| parachute | 0.2479 | 0.2453 | −0.0026 |

*Table 13.* **Effect of mapping concepts on semantically related retention.** CLIP score is the mean cosine similarity between generated images and a curated set of related prompts (e.g., other fish species for tench, other spaniel breeds for English springer). Mapping to an appropriate target preserves related concepts better than mapping to an empty prompt ($\Delta < 0$ in all cases). The effect is strongest for concepts with rich semantic neighborhoods (French horn: −0.037, cassette player: −0.024).

*Figure 10.* **Mapping target vs. empty target on Imagenette.** Each row shows one erased class; columns are random seeds. *Left:* mapping each class to a safe superordinate (e.g., *cassette player → electronic device*). *Right:* mapping to an empty prompt. Empty-target erasure collapses generations to off-distribution textures, while mapping-target erasure produces well-formed images of the safe target.

*Figure 11.* **Neighbor-probe ablation, "a chain saw".** The top row is the erased concept; the four rows below are semantically adjacent tools that should be *retained*. *Left:* mapping target. *Right:* empty mapping target. Under empty-target erasure, neighbors (especially *hand saw* and *power drill*) lose their characteristic appearance and drift toward off-distribution textures; under mapping-target erasure they remain visually intact.

**With mapping target**   **Without mapping target**

*Figure 12.* **Neighbor-probe ablation, "a golf ball".** Same layout as Figure 11. With an empty mapping target the suppression leaks into "a golf club" (which loses its metallic club-head and shifts toward generic landscape imagery); with a semantically appropriate mapping target the neighbors are preserved.

