# OpenReview forum: "UnHype: CLIP-Guided Hypernetworks for Dynamic LoRA Unlearning"
_ICML.cc/2026/Conference — ICML 2026 regular_

### Official Review · Reviewer_H72E · 2026-03-11

**Soundness:** 3
**Presentation:** 2
**Significance:** 3
**Originality:** 3
**Overall Recommendation:** 4
**Confidence:** 4

**Summary:**

This paper proposes a hypernetwork-based LoRA framework for concept unlearning in t2i diffusion models (UnHype). The main idea is to replace the standard practice of training and storing a separate LoRA adapter for each concept with a single shared hypernetwork.
UnHype learns a trajectory over LoRA weights using a gradient-matching objective, along with a retention term to preserve irrelevant concepts. The method is evaluated on several unlearning settings using two types of models, and achieves competitive results compared with baselines.

**Compliance With Llm Reviewing Policy:**

Affirmed.

**Final Justification:**

The authors have adequately addressed my major concerns. The additional ablation studies clarify the contribution of each design component, and the expanded multi-concept experiments provide stronger support for the scalability claim. The comparisons with updated baselines and the discussion on robustness also help consolidate the paper's contributions.

**Key Questions For Authors:**

Please see Weaknesses1-5.

I'd be happy to raise the score if most of the problems are resolved.

**Limitations:**

Yes

**Strengths And Weaknesses:**

**Strengths**:

1. This paper addresses the parameter management and scalability issues faced by LoRA-based unlearning methods in multi-concept scenarios by proposing a hypernetwork to generate LoRA weights. The motivation is reasonable and has practical significance.

2. The overall method flow is clear and intuitive, and the core design is easy to understand. The framework is easy to transfer to diffusion models with different backbones.

3. From an engineering implementation perspective, this paper's approach is attractive: learning LoRA parameter trajectories, rather than training and storing a separate LoRA for each concept.

**Weaknesses**

1. Since one of the core claims of this paper is its multi-concept forgetting capability, a more convincing evaluation would require a larger object-level benchmark, such as Imagenette [1] with all 10 classes. Current object experiments only cover a small number of categories, making it difficult to fully support its claim of multi-concept scalability. The authors could also refer to larger-scale datasets [2] to further analyze the method's limits and trade-offs.

2. I am concerned that the conflicts between multiple concepts are not truly resolved, but merely shifted from the original LoRA adapter level to the shared parameter space of the hypernetwork. Therefore, a larger-scale multi-object unlearning experiment would more directly reflect whether the method truly mitigates concept interference.

3. This paper lacks key ablation studies to validate the necessity of each design choice, making it difficult to determine which component is actually responsible for the observed performance improvement.

4. The input to the hypernetwork is CLIP embedding, and this type of semantic representation is inherently susceptible to adversarial attacks. This paper does not provide relevant experiments on robustness, so it is unclear whether the method remains stable and reliable under adversarial inputs or semantic perturbations.

5. Comparing the methods presented in this paper with the updated baseline [3,4] can enhance the persuasiveness.

Given the time constraints, a large number of experiments are not required, but at least it should be possible to further demonstrate how to address the above concerns.

[1] Jeremy Howard et al. Imagenette: A smaller subset of imagenet for quick experiments.

[2] Deng, Kaiyuan, et al. "Forget Many, Forget Right: Scalable and Precise Concept Unlearning in Diffusion Models." arXiv preprint arXiv:2601.06162 (2026).

[3] Li, Gen, et al. "Sculpting memory: Multi-concept forgetting in diffusion models via dynamic mask and concept-aware optimization." Proceedings of the IEEE/CVF International Conference on Computer Vision. 2025.

[4] Gong, Chao, et al. "Reliable and efficient concept erasure of text-to-image diffusion models." European Conference on Computer Vision

---

> ### Author Rebuttal · Authors · 2026-03-30
>
> We are greatly thankful to the Reviewer H72E for the recognition of the value of our work as well as for the constructive suggestions for additional experiments.
>
> ## Additional Experiment: Imagenette (10-Concept Removal)
> As requested, we conducted an additional experiment on the Imagenette dataset in the most challenging 10-concept removal setting. As shown in the table below, UnHype achieves the lowest total accuracy, indicating more effective forgetting, while maintaining the highest CLIP score, demonstrating strong alignment between prompts and generated images. UnHype remains competitive even compared to recent baselines. Furthermore, we have already demonstrated the scalability of our approach to multi-concept scenarios with the 100-celebrity removal task, as reported in Table 4 of the paper.
>
> | Method | tench | springer | cassette | chainsaw | church | horn | truck | pump | golf | parachute | Total Acc ↓ | CLIP ↑ |
> |--------|-------|----------|----------|----------|--------|------|-------|------|------|-----------|-------------|--------|
> | FMN | 0.75 | 0.96 | 0.23 | 0.64 | 0.74 | 1.00 | 0.91 | 0.80 | 0.95 | 0.91 | 0.789 | 29.87 |
> | AC | 0.14 | 0.96 | 0.11 | 0.83 | 0.89 | 0.96 | 0.54 | 0.62 | 0.53 | 0.49 | 0.607 | 29.32 |
> | ESD-x | 0 | 0.26 | 0.06 | 0.12 | 0.65 | 0.36 | 0.62 | 0.53 | 0.34 | 0.03 | 0.297 | 25.04 |
> | ESD-u | 0 | 0 | 0 | 0 | 0 | 0 | 0 | 0 | 0 | 0 | 0 | 22.52 |
> | SalUn | 0.92 | 0.01 | 0.34 | 0.07 | 0.01 | 0.09 | 0.09 | 0.58 | 0.05 | 0.10 | 0.226 | 25.37 |
> | MACE | 0.81 | 0.94 | 0.20 | 0.76 | 0.79 | 0.99 | 0.88 | 0.79 | 0.99 | 0.16 | 0.732 | 29.62 |
> | SPM | 0.65 | 0.70 | 0.00 | 0.32 | 0.77 | 0.27 | 0.62 | 0.29 | 1.00 | 0.67 | 0.529 | 29.31 |
> | UCE | 0.051 | 0.009 | 0.022 | 0.048 | 0.196 | 0.037 | 0.287 | 0.053 | 0.077 | 0.083 | 0.085 | 29.45 |
> | RECE | 0.010 | 0.020 | 0.011 | 0.031 | 0.134 | 0.021 | 0.153 | 0.042 | 0.042 | 0.030 | 0.049 | 29.27 |
> | SP | 0.01 | 0.00 | 0.05 | 0.03 | 0.17 | 0.00 | 0.41 | 0.05 | 0.12 | 0.00 | 0.084 | 26.43 |
> | **UnHype** | 0.085 | 0.000 | 0.000 | 0.000 | 0.005 | 0.010 | 0.115 | 0.025 | 0.030 | 0.035 | **0.031** | **30.61** |
>
> ## Additional Experiment: ImageNet-Confuse5
>
> We also conduct experiments on the ImageNet-Confuse5 benchmark proposed in [5]. This benchmark involves erasing five groups of visually similar concepts from ImageNet, with each group containing a pair of concepts to be removed. The results further confirm that UnHype can achieve fine-grained disentanglement while effectively avoiding concept confusion. Please note that due to the space limitations, we don't present every class below.
>
> | ImageNet-Confuse5 classes | SD | FMN | SPM | ESD | MACE | UCE | RECE | SP | ScaPre | **UnHype** |
> |----------------|---------|-----|-----|-----|------|-----|------|----|--------|------------|
> | *golden retriever* | 90.0 | 82.0 | 84.0 | 62.0 | 83.0 | 5.0 | 5.3 | 61.5 | 7.5 | 2.5 |
> | *labrador retriever* | 80.8 | 74.8 | 74.6 | 56.8 | 73.6 | 5.8 | 6.1 | 56.1 | 5.0 | 15.0 |
> | german shepherd | 78.3 | 71.3 | 71.0 | 49.3 | 69.9 | 3.3 | 3.1 | 48.7 | 76.8 | 100.0 |
> | Chesapeake Bay retriever | 93.3 | 85.3 | 87.3 | 67.3 | 86.3 | 8.3 | 8.0 | 66.8 | 89.2 | 16.0 |
> | pug | 90.0 | 84.0 | 83.8 | 63.0 | 82.8 | 6.7 | 6.4 | 62.3 | 83.4 | 100.0 |
> | *tabby* | 86.7 | 79.7 | 81.6 | 58.7 | 80.7 | 11.7 | 12.0 | 58.0 | 23.3 | 25.5 |
> | *tiger cat* | 80.0 | 72.0 | 71.8 | 53.0 | 70.8 | 5.0 | 5.2 | 52.4 | 9.2 | 30.5 |
> | Persian cat | 85.0 | 78.0 | 80.2 | 56.0 | 79.2 | 3.3 | 3.1 | 55.4 | 80.2 | 99.5 |
> | Siamese cat | 79.2 | 72.2 | 72.0 | 52.2 | 71.0 | 4.2 | 4.0 | 51.6 | 76.2 | 98.5 |
> | Granny Smith | 93.3 | 85.3 | 85.1 | 63.3 | 83.9 | 1.7 | 1.5 | 62.7 | 76.2 | 96.5 |
> | *yawl* | 74.2 | 66.2 | 68.4 | 44.2 | 67.4 | 0.0 | 0.1 | 43.6 | 4.2 | 0.0 |
> | *lifeboat* | 84.2 | 77.2 | 77.0 | 55.2 | 75.8 | 0.0 | 0.1 | 54.6 | 2.5 | 2.5 |
> | catamaran | 80.8 | 72.8 | 75.1 | 50.8 | 74.0 | 5.8 | 6.0 | 50.2 | 77.4 | 91.0 |
> | schooner | 81.7 | 73.7 | 76.0 | 51.7 | 74.9 | 10.0 | 10.3 | 51.1 | 78.3 | 80.0 |
> | *soccer ball* | 85.0 | 77.0 | 79.2 | 55.0 | 78.2 | 1.7 | 1.9 | 54.4 | 2.5 | 5.0 |
> | *volleyball* | 84.2 | 76.2 | 76.0 | 55.2 | 74.8 | 0.0 | 0.1 | 54.6 | 0.0 | 0.0 |
> | rugby ball | 92.5 | 84.5 | 86.8 | 62.5 | 85.7 | 9.2 | 9.0 | 61.9 | 71.7 | 50.5 |
> | ping-pong ball | 94.2 | 86.2 | 85.9 | 64.2 | 84.8 | 25.8 | 26.0 | 63.7 | 60.8 | 81.5 |
> | **Unlearn Acc (↓)** | 83.9 | 76.5 | 77.5 | 55.6 | 76.4 | **2.9** | 3.1 | 55.0 | 5.8 | 9.0 |
> | **Preserve Acc (↑)** | **86.6** | 78.9 | 79.7 | 57.7 | 78.6 | 5.6 | 5.5 | 57.1 | 76.3 | 83.3 |
> | **Overall Acc (↑)** | 27.2 | 36.2 | 35.1 | 50.2 | 36.3 | 10.6 | 10.4 | 50.3 | 84.3 | **87.0** |
> | **CLIP (↑)** | **31.43** | 30.45 | 30.60 | 29.78 | 30.98 | 28.04 | 27.23 | 29.84 | 30.15 | 30.60 |
>
> *Italic = erased class.*
> ## Ablations and Adversarial Attacks:
> Please refer to our responses to Reviewers wWAN and Ejkb
>
> [5] Kaiyuan Deng, et al. "Forget Many, Forget Right: Scalable and Precise Concept Unlearning in Diffusion Models." arXiv preprint arXiv:2601.06162 (2026).

---

> > ### Author Rebuttal · Reviewer_H72E · 2026-04-01
> >
> > The authors have adequately addressed my major concerns. The additional ablation studies clarify the contribution of each design component, and the expanded multi-concept experiments provide stronger support for the scalability claim. The comparisons with updated baselines and the discussion on robustness also help consolidate the paper's contributions.

---

> > > ### Author Response · Authors · 2026-04-05
> > >
> > > We kindly thank Reviewer H72E for the thorough and constructive review, and for taking the time to carefully evaluate our rebuttal. We are glad that our responses and additional experiments have fully addressed your concerns, and we truly appreciate the updated assessment.

---

### Official Review · Reviewer_Ejkb · 2026-03-12

**Soundness:** 2
**Presentation:** 1
**Significance:** 2
**Originality:** 2
**Overall Recommendation:** 4
**Confidence:** 4

**Summary:**

The paper proposes UnHype, a machine unlearning framework for image diffusion models that uses a hypernetwork to dynamically generate LoRA adapter weights conditioned on CLIP text embeddings. Rather than training a separate LoRA module for each concept to be erased, the hypernetwork learns to map any input concept to appropriate LoRA parameters in a single forward pass, drawing inspiration from Hypernet Fields that supervises the optimization trajectory rather than pre-computed target weights. The training objective combines a removal loss (which aligns the hypernetwork's predicted weight updates with the gradients of an unlearning task loss) and a retention loss (which pushes outputs toward zero for non-target concepts to prevent catastrophic forgetting). The authors evaluate on object erasure, nudity suppression, and simultaneous erasure of 100 celebrities, showing competitive results on both Stable Diffusion and Flux architectures while offering better generalization to synonyms than existing baselines.

**Compliance With Llm Reviewing Policy:**

Affirmed.

**Final Justification:**

UnHype introduces a novel application of hypernetworks to concept erasure in diffusion models, combining CLIP-conditioned weight generation with the Hypernet Fields trajectory-matching principle. The evaluation is broad (two architectures, three task types) and results are competitive, particularly for nudity suppression and synonym generalization. My primary concerns were all addressed in the rebuttal, and the authors have committed to concrete textual revisions on the scope of the efficiency claim to the multi-concept regime. They also reframed UnHype's motivation as primarily architectural (semantic switch behavior, CLIP-based generalization), which I find more accurate and compelling. With these promised revisions, I am raising my score to 4: Weak accept.

**Key Questions For Authors:**

- [Q1] **Compute cost comparisons.** Could you share a comparisons of UnHype vs. baselines in terms of training time, inference time, and GPU memory? The paper mentions that UnHype trains in ~3 hours for nudity erasure vs. 24+ hours for SAeUron, but a systematic comparison across all tasks and methods would be valuable. In particular, what is the inference overhead of the hypernetwork forward pass relative to standard LoRA application?
- [Q2] **Source of task loss gradients.** The removal loss requires computing $\nabla\_{\theta_s} \mathcal{L}\_{task}$, which is the gradient of the task loss w.r.t the current LoRA weights $\theta_s = H_\phi(c,s)$. Does this mean that at each training step, you must run a forward step through the diffusion model with the LoRA weights $\theta_s$ applied, and backpropagate the task loss to obtain the gradients? This appears computationally similar to just performing a LoRA fine-tuning step, which makes it unclear whether UnHype truly resolves the ``per-concept bottleneck'' mentioned in Lines 240-245. UnHype may still have the advantage that the LoRA weights are not stored in memory, but the runtime costs may still be comparable. Could the authors clarify this distinction?
- [Q3] **Generalization beyond training concepts.** The paper claims that conditioning on CLIP embeddings enables generalization to synonyms not seen during training (e.g., "warbler" generalizing from "bird"). How far does this generalization extend? Is this generally conditioned on the generalizability of CLIP? If the hypernetwork is trained to erase "nudity" and related terms, does it also suppress adversarial prompts that are semantically similar but lexically distant? Some evaluation of robustness to prompt paraphrasing would be valuable.

**Limitations:**

Yes

**Strengths And Weaknesses:**

## Strengths
- [S1] **Strong originality.** To the best of my knowledge, applying hypernetworks and Hypernet Fields to machine unlearning is a genuinely novel idea. The key insight of modeling the entire optimization trajectory of LoRA weight updates rather than learning a direct mapping to final weights is a fresh perspective that could lead to follow-up work in the intersection between unlearning and parameter-efficient adaptation.
- [S2] **Important and practical problem.** Concept erasure in diffusion models is a timely problem with clear real-world relevance (e.g., NSFW suppression, identity removal). The paper tackles a real issue in scalability as well: separate fine-tuning runs per concept can become prohibitive when many concepts need to be erased simultaneously. The demonstration of erasing 100 celebrities with a single model is a compelling use case that highlights this advantage.
- [S3] **Wide breadth of evaluation.**  The paper evaluates across two architectures (Stable Diffusion and Flux), three task types (object, nudity, celebrity erasure), and multiple established benchmarks (I2P, NudeNet, GIPHY Celebrity Detector, MS-COCO FID/CLIP). The results are generally competitive and the inclusion of both efficacy and retention metrics gives a fairly comprehensive picture.

## Weaknesses
- [W1] **Weak presentation.** The writing could be tightened in several places. For instance, the introduction enumerates limitations of existing LoRA-based methods (limited semantic adaptability, static structure, scalability issues) but does not ground these claims with concrete examples or empirical evidence. It would strengthen the paper to show, even briefly, a failure case of a static LoRA approach that UnHype resolves would make the motivation more tangible. Additionally, the relationship between the task loss (adapted from UnGuide) and the removal loss (from Hypernet Fields) could be explained more intuitively; currently the reader must piece together how these two components interact during training.
- [W2] **Missing key ablations.** The experimental section demonstrates that UnHype outperforms baselines on aggregate metrics, but provides limited insight into why it works and which design choices matter. This makes it difficult to assess the robustness of the approach or to guide practitioners in adapting it to new settings. Specifically, the following ablations could strengthen the paper:
  1. **Necessity of mapping concepts.** The task loss steers the model away from a forget concept $c$ and towards a mapping concept $c_m$. Is this mapping essential, or would steering toward the unconditional prediction suffice? that is, what happens if we use $\epsilon_{target} = \epsilon_{\theta^\*}(z_t, t) - \gamma(\epsilon_{\theta^\*}(z_t, t, c) - \epsilon_{\theta^*}(z_t, t))$ as the steered target prediction instead without any mapped concepts?
  2. **Loss weighting sensitivity.** The final objective is $\mathcal{L}\_{final} = \lambda\_{remove} \cdot \mathcal{L}\_{remove} + \lambda\_{retain} \cdot \mathcal{L}\_{retain}$, but the paper does not report what values were used or how sensitive performance is to this balance. A sweep over different ratios and discussion on (1) when retention begins to degrade and (2) when erasure becomes incomplete would be informative.
  3. **Trajectory length $S$.** The hypernetwork models an optimization trajectory over $s \in [0,S]$, and $S$ is described as a fixed hyperparameter. How sensitive are the results to this choice? Too small an $S$ could yield insufficient unlearning, but if too large, it may cause instability.
  4. **Number of retain concepts.** During training, retain concepts are sampled to enforce the semantic switch behavior. How many are needed? If erasing one CIFAR-10 class, must all nine remaining classes be used for retention, or does a smaller subset suffice? This has direct implications for the practical cost of assembling training data.
- [W3] **Limited analysis of failure modes.** It would be interesting to see what cases UnHype finds difficult. For example: Are there concepts that are particularly difficult to erase (e.g., highly polysemous words)? Does the hypernetwork ever produce non-negligible LoRA weights for retain concepts, and if so, what is the downstream effect? Understanding the boundaries of the method would be important in assessing its robustness.

---

> ### Author Rebuttal · Authors · 2026-03-30
>
> We kindly thank Reviewer Ejkb for their helpful and precise comments.
> We provide additional figures in: https://anonymous.4open.science/r/icml-rebuttal-images-2835/icml-rebuttal-images.pdf
> ## Training Time Analysis
> The table below reports the training time compared to baseline methods, measured on a single NVIDIA RTX 4090 GPU. While UnHype requires a similar amount of training time as AdvUnlearn, we emphasize that only our method enables efficient scaling from single- to 100-concept removal (e.g., the celebrity benchmark) within the same time budget.
> | Erasure Methods | Total Time (mins) |
> |-----------------|-------------------|
> | ESD | 41.27 |
> | RACE | 113.17 |
> | RECE | 0.38 |
> | AdvUnlearn | 146.62 |
> | STEREO | 41.80 |
> | UnHype | 148.00 |
>
> We also measure inference time comparing SD with a LoRA adapter and with the UnHype hypernetwork, demonstrating that the overhead difference is negligible.
>
> | Method | Mean (s) | Std (s) | Overhead |
> |---------------|----------|---------|-----------|
> | SD | 2.543 | 0.059 | +0.0% |
> | SD + LoRA | 2.659 | 0.003 | +4.6% |
> | SD + UnHype | 2.690 | 0.005 | +5.8% |
> ## Source of Task Loss Gradients
> Each training step involves a forward and backward pass, similar in cost to a single LoRA fine-tuning step. The difference is that UnHype trains one model for all concepts jointly, whereas per-concept LoRA methods (e.g., MACE) require N separate runs. For 100-celebrity removal, this means one ~3‑hour run versus 100 20-minute LoRA trainings, each needing hyperparameter tuning and checkpointing. Beyond efficiency, the hypernetwork architecture provides qualitative advantages: by generating concept-conditioned LoRA weights from a shared parameter space, it reduces mutual interference between concepts compared to concatenated adapters, and enables generalization to unseen paraphrases. We support this empirically through the nudity removal results, where UnHype produces mapped images most visually similar to the originals (see Fig. 1 in the paper).
> ## Generalization Beyond Training Concepts / Retain Concepts
> We would like to clarify that the CIFAR-10 class removal evaluation strictly follows the protocol from MACE, which treats the remaining nine classes as unknown to the model during the removal process and measures the model’s ability to preserve them via specificity. For this task, we use CIFAR-100 as the retain dataset, which contains no CIFAR-10 classes.
> We evaluate generalization by testing on synonyms of CIFAR-10 classes that were never seen during training. The I2P benchmark includes prompts associated with the concept of “nudity” not through lexical similarity, but via visual alignment (e.g., “cyborgs by Edouard Manet”). Additionally, our method shows state-of-the-art robustness to paraphrasing and compositional variations under two adversarial attacks (see our response to Reviewer wWAN).
> ## Missing Key Ablations
> Due to the space limitations, we present the requested ablations in the response to the Reviewer wWAN, please refer to them.
> ## Limited analysis of failure modes.
> In Figure 4 in the anonymised repo, we present several failure cases from the nudity removal task. Due to the continuous nature of the hypernetwork and LoRA, concepts that are related to nudity only at the textual level may be partially mapped to a “fully dressed person.” This can lead to unintended artifacts, such as the appearance of clothing on objects or animals.
> ## Necessity of Mapping Concepts
> We conduct an additional experiment to demonstrate the importance of mapping concepts, using a 10-class removal setting on Imagenette. In this setup, the removed objects are mapped to a neutral prompt rather than to a specific, arbitrary target concept. Although the quantitative results suggest that selecting mapping concepts does not significantly improve removal accuracy, we observe unintended side effects: concepts closely related to the target (e.g., “a golf club” when removing “a golf ball”) are adversely affected when the removed concept is mapped to an empty prompt.
> We support this observation both qualitatively (see Fig. 2 in the anonymised repo, highlighted in red boxes) and through a more detailed evaluation based on CLIP scores for semantically related prompts that should be retained. Notably, the standard evaluation protocol for Imagenette removal considers only retention accuracy measured using COCO CLIP, and does not account for effects on semantically related concepts.
> | erased_concept | mean_clip_mapping | mean_clip_empty | mean_delta |
> |--------|-------|----------|----------|
> | tench | 0.2590 | 0.2546 | -0.0044 |
> | English springer | 0.2615 | 0.2399 | -0.0216 |
> | cassette player | 0.2265 | 0.2026 | -0.0239 |
> | chain saw | 0.2424 | 0.2203 | -0.0221 |
> | church | 0.2361 | 0.2322 | -0.0039 |
> | French horn | 0.2466 | 0.2097 | -0.0369 |
> | garbage truck | 0.2482 | 0.2384 | -0.0097 |
> | gas pump | 0.2331 | 0.2172 | -0.0159 |
> | golf ball | 0.2538 | 0.2361 | -0.0177 |
> | parachute | 0.2479 | 0.2453 | -0.0026 |

---

> > ### Author Rebuttal · Reviewer_Ejkb · 2026-04-03
> >
> > Thank you authors for the thorough and detailed rebuttal. The additional experiments are very much appreciated (particularly the mapping concept ablations and computational overhead measurements) and have addressed many of my concerns.
> >
> > However, I still wish to flag one remaining issue regarding [Q2] (Source of Task Loss Gradients). The authors confirm that each training step requires a forward and backward pass through the diffusion model, comparable in cost to a single LoRA fine-tuning step. The primary efficiency argument then rests on amortization: one ~3-hour training run replaces 100 separate per-concept LoRA runs in the 100-celebrities setup. While this is a valid practical advantage for multi-concept erasure, it is a narrower claim than what the paper seems to convey. Several passages in the introduction and Section 4 frame per-concept LoRA methods as fundamentally limited by computational cost and suggest that UnHype resolves this bottleneck in a general sense. Since the per-step cost is in fact comparable, we would ask the authors to revise these claims to more precisely scope the efficiency advantage to the multi-concept amortization setting, and to be transparent about the fact that for single- or few-concept erasure, UnHype does not offer a meaningful training-time reduction over standard LoRA fine-tuning.
> >
> > Conditioned on this revision, I would be willing to raise the score to a 4: Weak accept.

---

> > > ### Author Response · Authors · 2026-04-05
> > >
> > > We thank Reviewer Ejkb for the constructive follow-up. We fully agree and commit to revising the paper in the camera ready. Below we outline the specific textual changes.
> > >
> > > ## **Proposed revision 1 — Introduction (paragraph 3)**.
> > >
> > > We will revise the three listed limitations of LoRA-based methods to:
> > >
> > > Despite their effectiveness, existing LoRA-based unlearning methods exhibit several key limitations. First, they apply a **global, static weight modification — once merged, the adapter affects every forward pass regardless of the input prompt**, which can lead to overly broad forgetting that degrades semantically adjacent concepts. Second, this rigid structure limits flexibility with context-dependent or compositional prompts. Third, **while the per-concept training cost of a single LoRA is modest**, scalability becomes a practical bottleneck when many concepts must be erased simultaneously, as each requires a separate run with its own hyperparameter tuning and checkpointing.
> > >
> > > ## **Proposed revision 2 — Introduction (contributions, bullet point 2)**.
> > >
> > > We will replace:
> > >
> > > Unlike previous methods that require separate fine-tuning for every target, this design supports simultaneous unlearning across multiple concepts, significantly lowering computational and memory costs.
> > >
> > > with:
> > >
> > > Unlike previous methods that require separate **LoRA** fine-tuning for every target, this design supports simultaneous unlearning across multiple concepts **within a single training run**, significantly lowering computational costs **in the multi-concept regime**.
> > >
> > > ## **Proposed revision 3 — Section 4, paragraph 2 (Lines 240–245)**.
> > >
> > > We will replace the sentence beginning with "Baseline per-concept fine-tuning approaches are computationally prohibitive…" with:
> > >
> > > While the per-step cost of LoRA fine-tuning is individually modest, per-concept approaches become a practical bottleneck in the multi-concept regime: erasing 100 celebrities requires 100 independent training runs with separate hyperparameter tuning and checkpointing. For single- or few-concept erasure, the training cost of UnHype is comparable to standard LoRA fine-tuning; the efficiency advantage manifests specifically through amortization over many concepts."
> > >
> > > ## **Proposed revision 4 — Section 5.4 (Celebrity Erasure), added sentence:**
> > >
> > > At the end of the paragraph we will add:
> > >
> > > The per-step cost of UnHype is comparable to standard LoRA fine-tuning (see the training time comparison in the appendix). The efficiency gain stems from amortization: a single run jointly handles all target concepts, whereas per-concept methods require N independent runs.
> > >
> > >
> > > ## Summary
> > >
> > > Beyond efficiency, we highlight that UnHype's motivation is primarily architectural. Standard LoRA applies a global, static modification — every forward pass is affected regardless of input. The hypernetwork instead generates concept-conditioned LoRA weights active only near target concepts, producing near-zero modifications otherwise. This "semantic switch" enables more surgical interventions that better preserve general capabilities. Moreover, by conditioning on CLIP embeddings, the hypernetwork generalizes to synonyms not seen during training (e.g., erasing "bird" also suppresses "warbler" and "owl"), whereas static LoRA adapters are bound to the exact concepts they were fine-tuned on.
> > > Regarding computational cost: while the removal loss requires a forward-backward pass through the diffusion model, the retention loss is a simple L2 penalty on the hypernetwork outputs and does not require a diffusion model pass. UnHype's total training time (~148 min) covers all target concepts in a single run — for 100-celebrity erasure, per-concept methods such as MACE require 100 independent runs, resulting in a substantially higher total cost. Furthermore, the hypernetwork generates weights from a shared parameter space, reducing mutual interference between concepts compared to merging independently trained adapters. At inference, the overhead is only ~30 ms per image, confirming no meaningful latency penalty.
> > > We believe these revisions precisely scope the efficiency claim to the multi-concept amortization setting, as requested. We will incorporate all changes in the revised version and are grateful for the reviewer's feedback.

---

### Official Review · Reviewer_wWAN · 2026-03-12

**Soundness:** 3
**Presentation:** 3
**Significance:** 3
**Originality:** 3
**Overall Recommendation:** 4
**Confidence:** 4

**Summary:**

This paper addresses the scalability bottleneck of concept-level machine unlearning in diffusion models. Standard LoRA-based methods require training and storing separate adapters for each concept, which often reduces generalization capability and leads to high storage costs. To mitigate this issue, the authors propose the UnHype framework, which amortizes the unlearning process by using a hypernetwork to dynamically generate concept-conditioned LoRA weights from semantic embeddings. UnHype consolidates concept removal into a single parameter generator by optimizing a joint objective that balances target concept suppression and non-target distribution preservation. The authors validate UnHype on object erasure, NSFW filtering, and identity removal tasks using Stable Diffusion and Flux architectures, achieving satisfactory results in both concept suppression and irrelevant content preservation.

**Compliance With Llm Reviewing Policy:**

Affirmed.

**Key Questions For Authors:**

1. Can you provide evaluations of the model's behavior under adversarial prompting, compositional prompts, or semantic paraphrasing? This would provide stronger evidence of genuine unlearning rather than superficial suppression.

2. In experiments leveraging stronger text encoders for UnHype, were the baseline methods granted access to these same embeddings? A controlled comparison is necessary to isolate the architectural contribution of the hypernetwork.

3. How sensitive is the framework to trajectory length and LoRA rank? Please provide ablation results to justify the rationale behind these specific design choices.

**Limitations:**

Yes.

**Strengths And Weaknesses:**

The method is technically sound and well-founded. By generating concept-specific LoRA weights through a hypernetwork, it constructs a logically coherent framework for concept removal, with optimization objectives that closely align with typical unlearning scenarios. The experimental evaluation covers multiple removal tasks across two mainstream architectures, Stable Diffusion and Flux, demonstrating the method's applicability. However, the evidence for genuine concept removal still requires strengthening. The current evaluation relies heavily on proxy metrics such as CLIP classifiers or external detectors, which struggle to definitively determine whether the target concept has been permanently eliminated under corresponding prompts or merely temporarily suppressed.

The paper features a clear structure and rigorous logical flow, with the research motivation, technical approach, and experimental design effectively conveying the core ideas. Nevertheless, certain implementation details remain insufficiently elaborated. The authors are encouraged to further clarify the hypernetwork architecture, the injection strategy for LoRA weights, and to provide in-depth analysis of how key hyperparameters—including trajectory length, hypernetwork capacity, and LoRA rank—impact model performance.

Achieving precise concept removal without full retraining represents a critical challenge in generative modeling, with direct implications for model safety, privacy protection, and governance. The proposed framework not only performs well on the targeted tasks but also offers new insights for efficient model editing in broader contexts. To enhance its practical value, the authors should validate the method's robustness under more diverse and challenging prompt conditions.

The core innovation lies in formulating concept-level unlearning as a hypernetwork-driven LoRA weight generation process, departing from the conventional approach of training separate adapters for each concept. While the individual technical components are relatively common, their integrated application in the context of concept removal demonstrates strong system-level design, endowing this work with distinctive methodological characteristics.

---

> ### Author Rebuttal · Authors · 2026-03-30
>
> We kindly thank Reviewer wWAN for the encouraging and constructive evaluation of our work.
> ## Robustness to Adversarial Attacks
> To evaluate the robustness of our method against textual adversarial attacks, we compare UnHype against state-of-the-art approaches specifically designed to mitigate such attacks. We consider both a black-box attack, Ring-A-Bell (RAB), and a white-box attack, UnlearnDiff (UD). The following table summarizes the robustness of UnHype relative to existing methods. Notably, UnHype is the only method that achieves top-tier adversarial robustness without degrading generation quality. While AdvUnlearn and STEREO [2] obtain comparable robustness scores, they do so at the cost of notably higher FID and lower CLIP scores.
> | Method | Erased (↓) | UD (↓) | RAB (↓) | FID (↓) | CLIP (↑) |
> | :--- | :--- | :--- | :--- | :--- | :--- |
> | SD 1.4 | 74.73 | 90.27 | 90.52 | 14.13 | 31.33 |
> | ESD | 3.15 | 43.15 | 35.79 | 14.49 | 31.32 |
> | AC | 1.05 | 25.80 | 89.47 | 14.13 | **31.37** |
> | UCE | 20.0 | 70.52 | 35.78 | 14.49 | 31.32 |
> | MACE | 6.31 | 41.93 | 5.26 | **13.42** | 29.41 |
> | RACE | 3.15 | 30.68 | 11.57 | 20.28 | 28.57 |
> | RECE | 4.21 | 53.08 | 9.47 | 14.90 | 30.94 |
> | AdvUnlearn | 1.05 | 3.40 | **0.00** | 15.84 | 29.27 |
> | STEREO [2] | 1.05 | 4.21 | 2.10 | 15.70 | 30.23 |
> | **UnHype** | **0.00** | **2.10** | 2.10 | 14.10 | 30.99 |
> ## Encoder Clarification
> We note that the stronger text encoder, used only in the 100 celebrities removal experiment (Table 4), is a frozen encoder producing embeddings for the hypernetwork. It is specific to our architecture and cannot be applied to other methods, being **distinct from the SD text encoder**. We apologize for the confusion. As shown, UnHype remains encoder-agnostic, supporting different encoders without modifying the core architecture. We hope to improve clarity in the camera-ready version.
> ## Core Ablation: Trajectory-Based Training
> The core ablation of our architecture is designed to justify the use of trajectory-based hypernetwork training. Following Hedlin et al. (2025), we evaluate performance under identical settings, but instead train the hypernetwork directly using the task-specific loss while keeping $T$ fixed:
> $$
> \phi \leftarrow \phi - \eta \nabla_{\phi} \, L_{\text{task}}\big(H_{\phi}(c, T)\, c\big)
> $$
> We report the quantitative results of this task-specific training ablation on the Imagenette [1] 10-class object removal task, observing a notable improvement in removed-object detection accuracy. A similar ablation, presented in Hedlin et al. (2025), was applied to human face image generation. It showed that the hypernetwork tended to overfit to the target, producing faces that closely resembled the reference—even when different personalized prompts were used. In contrast, our experiments show less pronounced quantitative effects, which we hypothesize is due to the remove prompts (and their textual augmentations) being mapped to a small set of general concepts.
>
> | Method      | tench | springer | cassette | chainsaw | church | horn | truck | pump | golf | parachute | Total Acc ↓ | CLIP ↑ |
> |-------------|-------|----------|----------|----------|--------|------|-------|------|------|-----------|-------------|--------|
> | Ours        | 0.255 | 0.005    | 0.000    | 0.030    | 0.040  | 0.025| 0.080 | 0.015| 0.050| 0.025     | **0.053**       | 0.306  |
> | Recon. loss | 0.125 | 0.005    | 0.005    | 0.010    | 0.090  | 0.100| 0.100 | 0.010| 0.130| 0.105     | 0.068       | 0.306  |
> ## Ablation on the LoRA Rank
> We conduct ablations to justify our hyperparameters, using 10-class removal from Imagenette [1], with total accuracy (lower = better) measuring ResNet50 performance on removed classes.
> Across ablations, rank 9 achieves the lowest total accuracy (0.0395) but performance remains stable across all tested ranks, trajectory length 300 hits a sweet spot between underfitting and overfitting the optimization trajectory, and a moderate loss weighting ratio of 1:1 proves most effective - all while CLIP scores remain virtually unchanged, confirming that these hyperparameters primarily affect erasure strength without degrading generation quality.
> | Rank | Total Acc ↓ | CLIP ↑ |
> |------|-------------|--------|
> | 1 | 0.0560 | 0.3061 |
> | 3 | 0.0795 | 0.3062 |
> | 6 | 0.0525 | 0.3062 |
> | 9 | 0.0395 | 0.3062 |
> ## Ablations on Trajectory Length
> | Hypersteps | Total Acc ↓ | CLIP ↑ |
> |------------|-------------|--------|
> | 100 | 0.0670 | 0.3061 |
> | 300 | 0.0405 | 0.3062 |
> | 500 | 0.0515 | 0.3062 |
> ## Ablation on Loss Weighting Ratio
> | RW | RW/Retain | Total Acc ↓ | CLIP ↑ |
> |----|-----------|-------------|--------|
> | 5 | 1:1 | 0.0305 | 0.3061 |
> | 10 | 2:1 | 0.0385 | 0.3061 |
> | 50 | 10:1 | 0.0510 | 0.3064 |
> | 100 | 20:1 | 0.0525 | 0.3062 |
>
> [1] Jeremy Howard et al. Imagenette: A smaller subset of imagenet for quick experiments.
> [2] Koushik Srivatsan, et al. "STEREO: Towards Adversarially Robust Concept Erasing from Text-to-Image Generation Models." CVPR 2025.

---

> > ### Author Rebuttal · Reviewer_wWAN · 2026-04-05
> >
> > Thank you for the authors' efforts, which have addressed most of my concerns. I will still keep my original score.

---

> > > ### Author Response · Authors · 2026-04-05
> > >
> > > We kindly thank Reviewer wWAN for their time and thoughtful engagement throughout the review process, and we truly appreciate your positive assessment of our work.
> > >
> > > If any points to clarify or follow-up questions remain, we would be happy to address them in the camera-ready version of the paper.

---

### Decision · Program_Chairs · 2026-04-30

**Decision:**

Accept (regular)

**Comment:**

This paper introduces a diffusion model unlearning framework using a shared hypernetwork that generates concept-aware LoRA weights on demand. The empirical study covers several representative unlearning tasks across multiple text-to-image backbones.

On the positive side, the use of a hypernetwork to model concept-conditioned LoRA updates is interesting in the unlearning setting. The empirical evaluation is reasonably broad, and the results suggest that the method achieves effective concept suppression while preserving overall generation quality.

For weaknesses, the original manuscript somewhat overstated the efficiency benefit. In addition, some conclusions rely on proxy metrics rather than direct evidence of complete concept removal.

The rebuttal addressed most major concerns by providing additional ablations, robustness results, and training/inference cost analysis. While some limitations still remain regarding stronger verification of true erasure and broader robustness analysis, the paper makes a solid and practically relevant contribution. Overall, I support acceptance.

For manuscript revision: The additional robustness evaluation and comparison with state-of-the-art robust unlearning methods are effortful and useful to address some common concerns from reviewers. Thus, please be sure to include those additional edvidences in the revised version.